# Advances in Temporal Point Processes: Bayesian, Neural, and LLM Approaches

**Feng Zhou**[1]                                         *feng.zhou@ruc.edu.cn*

**Quyu Kong**[2]                                      *kongquyu@gmail.com*

**Jie Qiao**[3]                                      *qiaojie.chn@gmail.com*

**Cheng Wan**[1]                                    *wancheng0256@ruc.edu.cn*

**Yixuan Zhang**[4]                                *zh1xuan@hotmail.com*

**Ruichu Cai**[3]                                  *cairuichu@gmail.com*

[1] *Center for Applied Statistics and School of Statistics, Renmin University of China, Beijing, China*
[2] *Independent Researcher*
[3] *School of Computer Science, Guangdong University of Technology, Guangzhou, China*
[4] *School of Statistics and Data Science, Southeast University, Nanjing, China*

**Reviewed on OpenReview:** *https://openreview.net/forum?id=SXgGKkShhT*

## Abstract

Temporal point processes (TPPs) are stochastic process models used to characterize event sequences occurring in continuous time. Traditional statistical TPPs have a long-standing history, with numerous models proposed and successfully applied across diverse domains. In recent years, advances in deep learning have spurred the development of neural TPPs, enabling greater flexibility and expressiveness in capturing complex temporal dynamics. The emergence of large language models (LLMs) has further sparked excitement, offering new possibilities for modeling and analyzing event sequences by leveraging their rich contextual understanding. This survey presents a comprehensive review of recent research on TPPs from three perspectives: Bayesian, deep learning, and LLM approaches. We begin with a review of the fundamental concepts of TPPs, followed by an in-depth discussion of model design and parameter estimation techniques in these three frameworks. We also revisit classic application areas of TPPs to highlight their practical relevance. Finally, we outline challenges and promising directions for future research.

## 1 Introduction

Many application scenarios generate time-stamped event sequences, which can be effectively modeled using temporal point processes (TPPs) (Daley & Vere-Jones, 2007). Examples include neural spike train data in neuroscience (Linderman & Adams, 2015), ask and bid orders in high-frequency financial trading (Bacry & Muzy, 2014), as well as tweets and retweets on social media (Kong et al., 2023). These event sequences, composed of asynchronous events, often influence one another and exhibit complex dynamics, making them more challenging to analyze compared to traditional synchronous time series problems. Investigating the

Table 1: Survey comparison.

| Survey | Years Covered | Frequentist TPP | Bayesian TPP | Neural TPP | LLM-based TPP | Training Methods | Applications |
|---|---|---|---|---|---|---|---|
| Yan (2019) | $\leq 2019$ | ✔ | ✗ | ✔ | ✗ | Limited | ✔ |
| Shchur et al. (2021) | $\leq 2021$ | ✔ | ✗ | ✔ | ✗ | Limited | ✔ |
| Hawkes (2018) | $\leq 2018$ | ✔ | ✗ | ✗ | ✗ | Limited | Finance |
| This survey | $\leq 2025$ | ✔ | ✔ | ✔ | ✔ | ✔ | ✔ |

underlying dynamic processes of such event sequences not only facilitates the prediction of future events but also helps uncover causal relationships.

In the statistics community, TPPs have a long-standing history of research, with numerous statistical TPP models proposed over the years. Examples include the classic Poisson process (Kingman, 1992), Hawkes process (Hawkes, 1971), and self-correcting process (Isham & Westcott, 1979), among others. Each of these models is particularly well-suited to specific applications. For instance, the Poisson process was used to model telephone call arrivals, while the Hawkes process, due to its ability to capture self-exciting characteristics, has been widely applied to model earthquakes and aftershocks.

Early TPP models are primarily parametric, requiring explicit specification of the parametric form of the model. However, this imposes limitations on their expressive power. To address these limitations, various nonparametric TPPs have been proposed within the statistics community, including approaches from both the frequentist and Bayesian frameworks, enabling more flexible modeling without the constraints of fixed parametric forms. In recent years, driven by rapid advancements in deep learning, the machine learning community has introduced approaches that combine neural network architectures with TPPs, referred to as neural TPPs. These models further enhance expressive power and are often more intuitive, simpler, and easier to train compared to statistical nonparametric TPPs. Over the past several years, the emergence of large language models (LLMs) has brought transformative changes to the field of artificial intelligence. With their rich contextual understanding and ability to process multimodal data, LLMs offer new possibilities for modeling event sequences. This paper reviews recent advances in TPPs based on Bayesian methods, deep learning, and LLMs, with a focus on model design and parameter estimation. Due to space limitations, we do not aim to cover every method in detail but instead emphasize fundamental principles and core ideas. We also revisit classic applications of TPPs and discuss key challenges and future research directions in the field. This taxonomy of recent advances in TPPs is illustrated in Figure 1. This survey is expected to give comprehensive background knowledge, research trends and technical insights for TPPs.

**Comparison with Existing TPP Surveys** Several surveys on TPPs in machine learning already exist, such as Yan (2019) and Shchur et al. (2021). The former summarizes advances in statistical and neural TPPs, while the latter provides a more detailed overview of neural TPPs. Additionally, surveys in other domains, such as finance, include Hawkes (2018). Compared with these works, this paper provides a comprehensive update. In the domain of statistical TPPs, we emphasize recent progress in Bayesian nonparametric TPPs, which has been largely overlooked, as most prior reviews (e.g., Yan (2019)) primarily focus on frequentist approaches. For neural TPPs, Shchur et al. (2021) covers works up to 2020, whereas this survey reviews advances from 2020 to 2025. Furthermore, we include a systematic review of the emerging area of LLM-based TPPs, which has gained significant attention in recent years but has not yet been comprehensively surveyed. A detailed comparison with existing surveys is provided in Table 1.

**Survey Methodology.** To construct this survey, we conducted a structured literature review covering major venues in machine learning, statistics, and related application domains. Specifically, we collected papers from top conferences and journals such as NeurIPS, ICML, ICLR, AISTATS, KDD, AAAI, JMLR, and IEEE/ACM Transactions, as well as relevant statistical journals. Our primary focus is on works published between 2020 and 2025, while also including earlier foundational studies for completeness. We include papers based on the following criteria: (i) the work proposes a novel TPP model or inference method, (ii) it introduces new training or estimation techniques applicable to TPPs, or (iii) it demonstrates significant applications or extensions (e.g., multimodal or LLM-based TPPs). The collected literature is then organized into three main categories—Bayesian, neural, and LLM-based approaches—based on modeling paradigms.

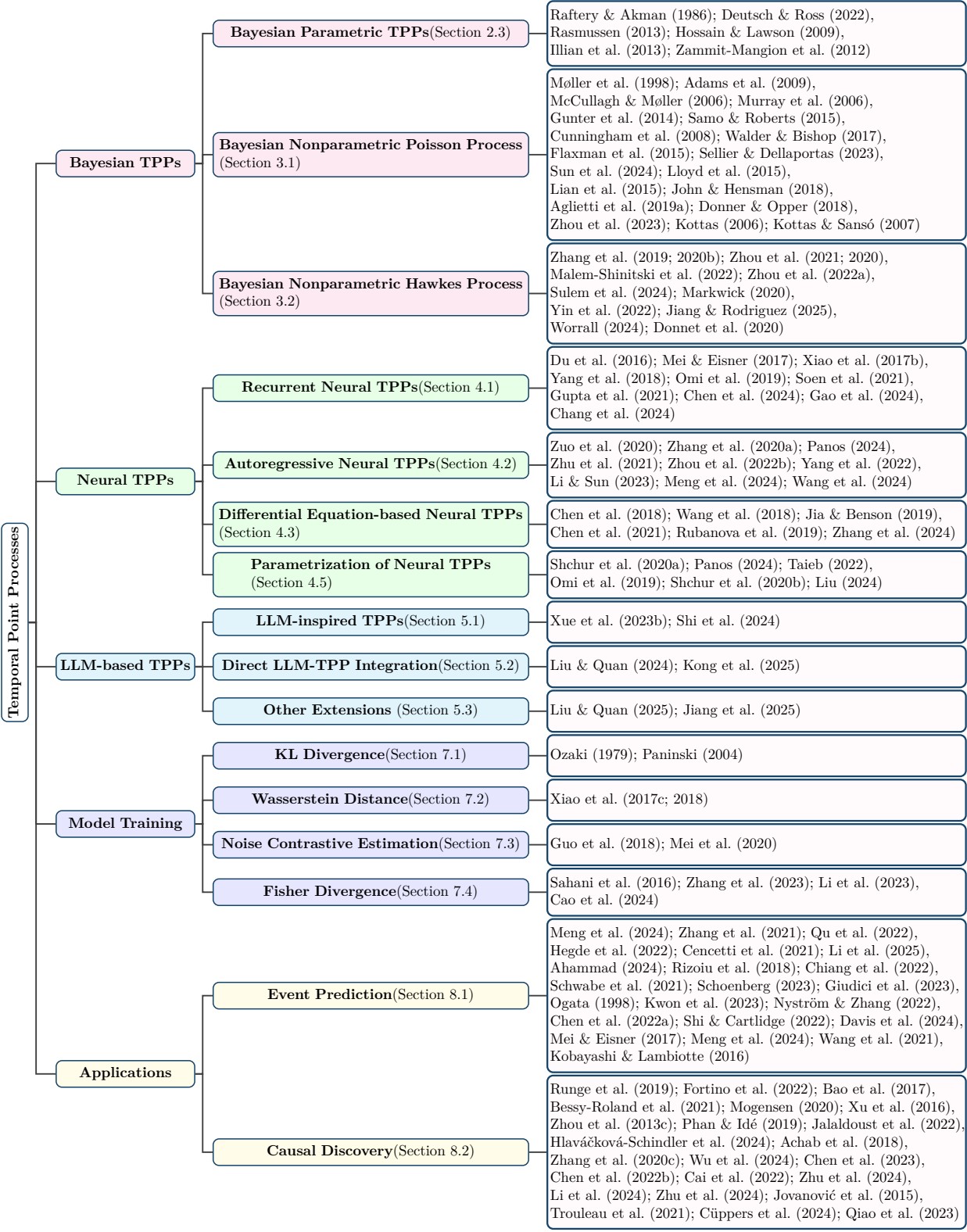

Figure 1: The taxonomy of Bayesian, neural, and LLM-based TPPs.

Due to space constraints, this survey does not aim to be exhaustive, but instead focuses on representative and influential works that highlight key methodological developments and research trends.

## 2 Background of TPPs

We first briefly review some of the core concepts of TPPs. For readers unfamiliar with TPPs, we recommend Rasmussen (2018) for a comprehensive introduction.

### 2.1 Unmarked TPP

A TPP is a stochastic process that models the occurrence of events over a time window $[0, T]$. A trajectory from a TPP can be represented as an ordered sequence $\mathcal{T} = (t_1, \ldots, t_N)$, $N(t) = \max\{n : t_n \leq t, t \in [0, T]\}$ represents the associated counting process. Here, $N$ denotes the random number of events within the interval $[0, T]$. TPPs can be defined using different parameterizations. One approach is to specify the distribution of the time intervals between consecutive events. We denote the $f(t_{n+1} \mid \mathcal{H}_{t_n})$ to be the conditional density function of $t_{n+1}$ given the history of previous events $t_1, \ldots, t_n$. In this work, $\mathcal{H}_{t^-}$ denotes the history of events up to but excluding time $t$, while $\mathcal{H}_t$ includes whether an event occurs at time $t$. The conditional density function sequentially specifies the distribution of all timestamps. Consequently, the joint distribution of all events can be factorized as:

$$f(t_1, \ldots, t_N) = \prod_{n=1}^{N} f(t_n \mid \mathcal{H}_{t_{n-1}}). \tag{1}$$

A TPP can be defined by specifying the distribution of time intervals. For example, a renewal process assumes that the time intervals are independent and identically distributed (i.i.d.), i.e., $f(t_n \mid \mathcal{H}_{t_{n-1}}) = g(t_n - t_{n-1})$, where $g$ is a probability density function defined on $(0, \infty)$. If we further specify $g(t_n - t_{n-1})$ to follow an exponential distribution, we obtain a homogeneous Poisson process, where each event occurs independently of the past.

The above approach can directly define some classic point process models. However, in general cases, event occurrences may depend on the entire history, making it less convenient to specify the model using the probability density function of time intervals. Instead, the conditional intensity function provides a more convenient way to describe how the occurrence of an event depends on its history. The conditional intensity function is defined as:

$$\lambda^*(t)dt = \frac{f(t \mid \mathcal{H}_{t_n})dt}{1 - F(t \mid \mathcal{H}_{t_n})} = \frac{P(t_{n+1} \in [t, t+dt] \mid \mathcal{H}_{t_n})}{P(t_{n+1} \notin (t_n, t) \mid \mathcal{H}_{t_n})} = P(t_{n+1} \in [t, t+dt] \mid t_{n+1} \notin (t_n, t), \mathcal{H}_{t_n})$$
$$= P(t_{n+1} \in [t, t+dt] \mid \mathcal{H}_{t^-}) = \mathbb{E}[N([t, t+dt]) \mid \mathcal{H}_{t^-}], \tag{2}$$

where $F(t \mid \mathcal{H}_{t_n}) = \int_{t_n}^{t} f(\tau \mid \mathcal{H}_{t_n})d\tau$ denotes the cumulative distribution function. Following tradition, we use * to indicate that the conditional intensity function is based on the history. The conditional intensity function has an intuitive interpretation: it specifies the average number of events in a time interval, conditional on the history up to but not including $t$. It is worth noting that the history $\mathcal{H}_{t_n}$ in the conditional density function differs from the history $\mathcal{H}_{t^-}$ in the conditional intensity function. This subtle distinction is often overlooked in many TPP works.

The conditional intensity function and the conditional density function are one-to-one[1]. This can be easily proven by inverting Equation (2) to express the conditional density function in terms of the conditional intensity function:

$$F(t \mid \mathcal{H}_{t_n}) = 1 - \exp\left(-\int_{t_n}^{t} \lambda^*(\tau)d\tau\right),$$
$$f(t \mid \mathcal{H}_{t_n}) = \lambda^*(t) \exp\left(-\int_{t_n}^{t} \lambda^*(\tau)d\tau\right). \tag{3}$$

This means we can define new TPP models by directly specifying a particular form of the conditional intensity function. For example, specifying a constant intensity defines a homogeneous Poisson process,

---

[1]The conditional intensity function must satisfy certain conditions.

while specifying a time-varying intensity function $\lambda^*(t) = \lambda(t)$ defines an inhomogeneous Poisson process. We can also define a Hawkes process by specifying a conditional intensity function:

$$\lambda^*(t) = \mu + \sum_{t_n < t} \phi(t - t_n), \tag{4}$$

where $\mu > 0$ is the baseline intensity, and $\phi(\cdot) : \mathbb{R}^+ \to \mathbb{R}^+$ is the triggering function[2]. The summation of influences from past events increases the likelihood of future events, making it suitable for modeling self-exciting effects. While many other forms of TPPs exist, we primarily focus on the Poisson process (history-independent) and the Hawkes process (history-dependent) in the following due to their widespread use. An illustration of the unmarked temporal point process, along with its conditional density function and conditional intensity function, is shown in Figure 2.

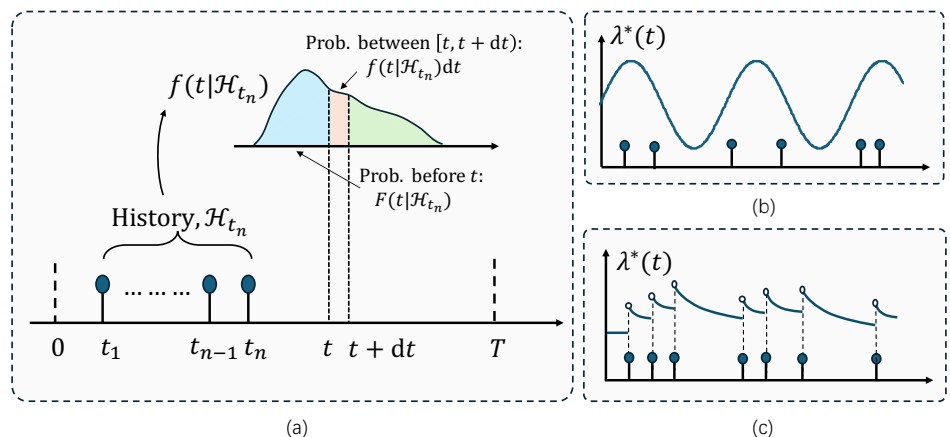

Figure 2: Illustration of the conditional density function and conditional intensity function in TPPs. (a) The conditional density function of the $(n+1)$-th event given the history $\{t_1, \ldots, t_n\}$; (b) the intensity function of an inhomogeneous Poisson process; (c) the conditional intensity function of a Hawkes process.

## 2.2 Marked TPPs

The above discussion focuses on the unmarked TPP, also known as the univariate TPP. However, TPPs can be extended to the marked case, represented as a time-ordered marked sequence $\mathcal{T} = ((t_1, k_1), \ldots, (t_N, k_N))$ over a time window $[0, T]$, where $k_n$ is the mark of the $n$-th event. The mark space can be continuous or discrete. In practice, discrete marks are more common, often referred to as multivariate TPPs. Similar to the unmarked case, marked TPPs can also be described using the conditional density function:

$$f((t_1, k_1), \ldots, (t_N, k_N)) = \prod_{n=1}^{N} f(t_n, k_n \mid \mathcal{H}_{t_{n-1}}), \tag{5}$$

where $f(t, k \mid \mathcal{H}_{t_n})$ is the joint density of time and mark, conditional on history. The history $\mathcal{H}_{t_n}$ now includes information about both the times and marks of past events.

Similarly, when event occurrences depend on the entire history, specifying the model using the probability density function becomes less convenient. In such cases, the conditional intensity function offers a more practical and expressive way to capture the dependence of events on historical information. The conditional intensity function is defined as:

$$\lambda^*(t, k) dt dk = \frac{f(t, k \mid \mathcal{H}_{t_n}) dt dk}{1 - F(t \mid \mathcal{H}_{t_n})} = \mathbb{E}[N(dt \times dk) \mid \mathcal{H}_{t-}],$$

---

[2]It requires $\int_0^\infty \phi(t)\, dt < 1$ to ensure the Hawkes process does not explode.

where $F(t \mid \mathcal{H}_{t_n}) = \int_{t_n}^{t} \int_{k} f(\tau, k \mid \mathcal{H}_{t_n}) dk d\tau$ is the conditional cumulative distribution function and the mark is marginalized out. The conditional intensity function specifies the average number of events with a mark $k$ in a time interval, conditional on the history up to but not including $t$.

We can also define a marked TPP by specifying a particular conditional intensity function. A classic example is the multivariate Hawkes processes, where the mark $k \in \{1, \ldots, K\}$ represents the event type. The conditional intensity function of multivariate Hawkes processes is given by:

$$\lambda^*(t, k) = \mu_k + \sum_{t_n < t} \phi_{k,k_n}(t - t_n), \tag{6}$$

where $\mu_k$ represents the baseline intensity for event type $k$, and $\phi_{k,k'}$ captures the triggering effects from events of type $k'$ on type $k$. A comparison of commonly used classical TPP models is provided in Table 2.

Table 2: Comparison of classic TPP models.

| Model | History dependence | Marks | Intensity form | Typical use |
|---|---|---|---|---|
| Poisson | No | No | $\lambda$ | Independent arrivals |
| inhomogeneous Poisson | No | No | $\lambda(t)$ | Time-varying rate |
| Hawkes | Yes | No | $\mu + \sum_{t_n < t} \phi(t - t_n)$ | Self-excitation |
| Multivariate Hawkes | Yes | Yes | $\mu_k + \sum_{t_n < t} \phi_{k,k_n}(t - t_n)$ | Type interaction |

## 2.3 Inference

There are various methods for estimating the parameters of TPPs. The most common method is maximum likelihood estimation (MLE). In this section, we introduce MLE and Bayesian inference for parametric TPPs. Assume we observe a trajectory of a marked TPP $\mathcal{T} = ((t_1, k_1), \ldots, (t_N, k_N))$ over the time window $[0, T]$. The unmarked case corresponds to the situation where there is only a single mark. Assume the marked TPP is specified by a parametric conditional intensity function $\lambda_\theta^*(t, k)$. Then, the likelihood function is given by:

$$f(\mathcal{T}; \theta) = \prod_{n=1}^{N} \lambda_\theta^*(t_n, k_n) \exp\left(-\int_0^T \lambda_\theta^*(t) dt\right), \tag{7}$$

where $\lambda_\theta^*(t) = \int \lambda_\theta^*(t, k) dk$ is the ground intensity. The proof of Equation (7) is straightforward. Simply substitute Equation (3) into Equation (1) to verify it for the unmarked case. The proof for the marked case follows a similar procedure. We can use numerical methods to maximize the log-likelihood to obtain parameter estimates.

MLE is a widely used parameter estimation method in the frequentist framework. It offers several advantages, such as consistency, asymptotic normality, and asymptotic efficiency. However, as a point estimation method, MLE cannot capture model uncertainty, which limits its applicability in high-stakes domains where understanding uncertainty is crucial. To address this issue, the Bayesian framework has been incorporated into TPPs (Raftery & Akman, 1986). In Bayesian TPPs, we impose suitable priors on the model parameters and then compute their corresponding posterior distribution, equipping the model with the ability to quantify uncertainty. Specifically, a Bayesian TPP is formally expressed as:

$$f(\theta \mid \mathcal{T}) = \frac{f(\mathcal{T} \mid \theta) f(\theta)}{\int f(\mathcal{T} \mid \theta) f(\theta) d\theta}, \tag{8}$$

where $f(\mathcal{T} \mid \theta)$ is the likelihood in Equation (7), $f(\theta)$ is the prior on model parameters, the denominator is the marginal likelihood, and $f(\theta \mid \mathcal{T})$ is the posterior distribution of model parameters. In general, the inference for Bayesian TPPs is more challenging than for frequentist TPPs because the TPP likelihood is not conjugate to any prior. This means that the posterior does not have an analytical expression and can only be obtained through approximation methods, such as Markov chain Monte Carlo (MCMC) (Deutsch & Ross, 2022; Rasmussen, 2013), variational inference (Lloyd et al., 2015; Zammit-Mangion et al., 2012), and Laplace approximation (Hossain & Lawson, 2009; Illian et al., 2013), among others.

# 3 Bayesian Nonparametric TPPs

Early work on TPPs was limited to parametric models, whether in the frequentist or Bayesian framework. These methods rely heavily on model assumptions, making them inflexible and often performing poorly on complex datasets. To address this limitation, many studies have proposed nonparametric methods. Yan (2019) provides a comprehensive description of frequentist nonparametric methods, while this paper focuses on Bayesian nonparametric TPPs, which not only enhance model flexibility but also incorporate the ability to quantify model uncertainty.

## 3.1 Bayesian Nonparametric Poisson Process

Bayesian nonparametric TPPs do not parameterize the intensity function into a fixed form. Instead, they treat the intensity function itself as a model parameter with infinite dimensions and impose suitable prior on it. For instance, in the case of an inhomogeneous Poisson process, the intensity function $\lambda(t)$ is treated as the parameter, and a prior $f(\lambda(t))$ is placed on it. The goal is then to compute the posterior distribution. This prior, being a distribution over functions, is commonly modeled using a Gaussian process (GP). Consequently, the Bayesian nonparametric framework is formulated as follows:

$$f(g(t) \mid \mathcal{T}) = \frac{f(\mathcal{T} \mid \lambda(t) = l \circ g(t))\mathcal{GP}(g(t))}{\int f(\mathcal{T} \mid \lambda(t) = l \circ g(t))\mathcal{GP}(g(t))dg}, \tag{9}$$

where $l(\cdot) : \mathbb{R} \to \mathbb{R}^+$ is a link function ensuring the intensity function is non-negative, e.g., exponential (log Gaussian Cox process (Møller et al., 1998)), scaled sigmoid (sigmoidal Gaussian Cox process (Adams et al., 2009)), square (permanental process (McCullagh & Møller, 2006)), etc. It is worth noting that computing Equation (9) is highly challenging, as the posterior is doubly intractable due to an intractable integral over $t$ in the numerator and another over $g$ in the denominator. This is a well-known problem in the field of Bayesian nonparametric TPPs (Murray et al., 2006).

Many methods have been proposed to solve Equation (9). Some studies focus on utilizing MCMC methods. Adams et al. (2009) proposed an MCMC inference method for a Poisson process with a sigmoidal GP prior. The core idea is to incorporate latent thinned points to make the posterior tractable. However, this method scales cubically with the number of data and thinned points. Gunter et al. (2014) extended this approach to multiple dependent Cox processes using multi-output Gaussian processes, and also derived an MCMC sampler for performing inference. Later, Samo & Roberts (2015) leveraged inducing points, a common technique in GP for reducing time complexity (Titsias, 2009), to derive an MCMC sampler that reduces the computational cost to linear w.r.t. the number of data points.

Some studies focus on methods based on the Laplace approximation. Flaxman et al. (2015) combined Kronecker methods with the Laplace approximation to enable scalable inference. Walder & Bishop (2017) proposed a fast Laplace approximation relying on the Mercer decomposition of the GP kernel. However, its tractability is limited to standard kernels such as the squared exponential kernel. To address this limitation, Sellier & Dellaportas (2023) introduced an alternative fast Laplace approximation leveraging the spectral representation of kernels. This approach retains tractability while accommodating a broader range of stationary kernels. Furthermore, Sun et al. (2024) extended this method to non-stationary kernels by leveraging sparse spectral representations to overcome the limitations of stationary kernels. This approach provides a low-rank approximation of the kernel, effectively reducing computational complexity from cubic to linear.

Another common approach is variational inference. Lloyd et al. (2015) introduced the first fully variational inference scheme for permanental processes. However, similar to Walder & Bishop (2017), its tractability is limited to certain standard types of kernels. Lian et al. (2015) further extended the method in Lloyd et al. (2015) to a multitask point process model, leveraging information from all tasks via a hierarchical GP. John & Hensman (2018) expanded the approach in Lloyd et al. (2015) to utilize the Fourier representation of the GP, enabling the use of more general stationary kernels. Aglietti et al. (2019a) presented a novel tractable representation of the likelihood through augmentation with a superposition of Poisson processes. This perspective enabled a structured variational approximation that captured dependencies across variables in the model. The method avoided discretization of the domain, did not require numerical integration over the input space, and was not limited to GPs with squared exponential kernels.

It is worth noting that, Donner & Opper (2018) introduced the data augmentation technique based on Pólya-Gamma variables to the field of Bayesian nonparametric TPPs. This technique is an improvement and extension of the method proposed by Adams et al. (2009). The method augments not only thinned points but also Pólya-Gamma latent variables for all data and thinned points in the likelihood. This enables the augmented likelihood to be conditionally conjugate to the GP prior. By leveraging the conditionally conjugacy, we can derive fully analytical Gibbs sampler, EM algorithm, and mean-field variational inference method. This method was later extended to jointly model multiple heterogeneous and correlated tasks—such as classification, regression, and point processes—using multi-output GPs. This extension facilitated information sharing across heterogeneous tasks while enabling nonparametric estimation (Zhou et al., 2023).

The works discussed above primarily use GP as prior. However, other forms of Bayesian nonparametric priors are also possible. For example, Kottas (2006); Kottas & Sansó (2007) used a Dirichlet process mixture of Beta densities as a prior for the normalized intensity function of a Poisson process. Bayesian nonparametric TPPs based on Dirichlet process mixtures and those based on GPs represent two orthogonal modeling paradigms, both capable of achieving Bayesian nonparametric inference. Inference for Dirichlet process mixture-based TPPs typically relies on specialized MCMC or variational inference techniques developed within the Dirichlet process framework. A detailed discussion of these methods is beyond the scope of this paper.

### 3.2 Bayesian Nonparametric Hawkes Process

For the Hawkes process, as shown in Equation (4), the conditional intensity function consists of the baseline intensity $\mu(\cdot)$[3] and the triggering function $\phi(\cdot)$. Therefore, the Bayesian nonparametric Hawkes process typically places GP priors on both $\mu(\cdot)$ and $\phi(\cdot)$ (we take the unmarked case as an example):

$$f(g(t), h(\tau) \mid \mathcal{T}) \propto f(\mathcal{T} \mid \mu(t) = l \circ g(t), \phi(\tau) = l \circ h(\tau)) \mathcal{GP}(g(t)) \mathcal{GP}(h(\tau)), \tag{10}$$

where $l(\cdot)$ is a link function ensuring $\mu(t)$ and $\phi(\tau)$ are non-negative, similar to Equation (9). Computing Equation (10) is more challenging than Equation (9) because in the likelihood of Hawkes process, $\mu(t)$ and $\phi(\tau)$ are coupled together, which significantly complicates the inference process.

To address this issue, a common approach is to augment a branching latent variable into the Hawkes likelihood to indicate whether each event is triggered by itself via the baseline intensity or by a previous event via the triggering function (Marsan & Lengline, 2008). The branching variable $\mathbf{X}$ is a lower triangular matrix with Bernoulli entries, where $x_{nm}$ indicates whether the $n$-th event is triggered by itself or a previous event $m$:

$$x_{nn} = \begin{cases} 1 & \text{if event } n \text{ is a background event,} \\ 0 & \text{otherwise,} \end{cases}$$

$$x_{nm} = \begin{cases} 1 & \text{if event } n \text{ is caused by event } m, \\ 0 & \text{otherwise.} \end{cases}$$

After augmenting the branching latent variable, the joint likelihood is expressed as:

$$f(\mathcal{T}, \mathbf{X} \mid \mu(t), \phi(\tau)) = \underbrace{\prod_{n=1}^{N} \mu(t_n)^{x_{nn}} \exp\left(-\int_0^T \mu(t)dt\right)}_{\text{baseline intensity part}} \cdot \underbrace{\prod_{n=2}^{N} \prod_{m=1}^{n-1} \phi(t_n - t_m)^{x_{nm}} \prod_{n=1}^{N} \exp\left(-\int_0^{T_\phi} \phi(\tau)d\tau\right)}_{\text{triggering function part}},$$

$$\tag{11}$$

where the support of triggering function is assumed to be $[0, T_\phi]$. If the branching variable $\mathbf{X}$ is marginalized out, we obtain the original likelihood.

It is clear that, after introducing the branching variable, the joint likelihood factorizes into two independent components: one corresponding to the baseline intensity and the other to the triggering function. These two components are linked through the branching variable $\mathbf{X}$. To the best of our knowledge, Marsan & Lengline (2008) was the first to identify this structure and subsequently proposed an EM algorithm that leverages

---

[3]Some studies treat the baseline intensity as a constant, but here we consider it as a more general function.

it: the E-step computes the posterior distribution of the branching variable, while the M-step estimates the parameters of both the baseline intensity and the triggering function. Later, Lewis & Mohler (2011) extended this approach to the frequentist nonparametric setting by treating the baseline intensity and the triggering function as two flexible, unconstrained functions. They imposed Good's roughness penalty (Goodd & Gaskins, 1971) on both functions and derived the solutions using the Euler–Lagrange equation. Zhou et al. (2013b) further extended the method in Lewis & Mohler (2011) to multivariate Hawkes processes by assuming that the triggering functions in the multivariate setting are linear combinations of a set of basis functions. Each basis function was then estimated using the Euler–Lagrange equation. There also exist frequentist nonparametric approaches that do not rely on the branching variable. For example, Bacry & Muzy (2016) proposed an estimation method based on solving a Wiener–Hopf equation that relates the triggering function to the second-order statistics. Additionally, Eichler et al. (2017); Reynaud-Bouret et al. (2010) attempted to estimate the triggering function by minimizing a quadratic contrast function using a grid-based representation of the triggering function. Bonnet & Sangnier (2024) extended the approach of Flaxman et al. (2017), which formulated Poisson process intensity estimation within a reproducing kernel Hilbert space (RKHS) framework, to the more complex setting of Hawkes processes.

In the Bayesian nonparametric Hawkes process setting, a similar strategy can be adopted. By introducing the branching variable, the Hawkes likelihood factorizes into two independent components, each of which can be viewed as an independent Poisson process. Since we place independent GP priors on these two components, we can directly apply the methods discussed in Section 3.1 to compute the posteriors of $\mu(t)$ and $\phi(\tau)$. This naturally leads to an iterative algorithm where, at each iteration, the posterior of $\mathbf{X}$ is used to update the posteriors of $\mu(t)$ and $\phi(\tau)$, which are then used to update the posterior of $\mathbf{X}$ in turn.

In recent years, many GP-based Bayesian nonparametric Hawkes process studies have adopted this iterative framework. Zhang et al. (2019) derived a Gibbs sampler and a maximum a posteriori (MAP) EM algorithm to estimate a nonparametric triggering function. Zhang et al. (2020b) extended the variational inference method from Lloyd et al. (2015) to the Hawkes process for estimating a nonparametric triggering function. Further, Zhou et al. (2021) applied variational inference to simultaneously estimate the nonparametric baseline intensity and triggering function. Zhou et al. (2020); Malem-Shinitski et al. (2022) extended the data augmentation method based on Pólya-Gamma variables from Donner & Opper (2018) to the Hawkes process. This work derived fully analytical Gibbs sampler, EM algorithm, and mean-field variational inference method. This approach was subsequently extended to nonlinear Hawkes processes to account for excitation and inhibition effects between interacting variables (Zhou et al., 2022a). Sulem et al. (2024) analyzed the theoretical properties of this approach; specifically, it provided concentration rates for the posterior distribution of the parameters under mild assumptions on both the prior and the model, and established consistency guarantees for the inferred Granger causal graph.

The works discussed above primarily use GP as prior. However, other forms of Bayesian nonparametric priors are also possible. For instance, Markwick (2020); Yin et al. (2022); Jiang & Rodriguez (2025) extended the Hawkes process to the Bayesian nonparametric setting using Dirichlet process mixture priors. Building on this line of research, Worrall (2024) proposed an online inference method using sequential Monte Carlo techniques. Additionally, Donnet et al. (2020) introduced priors based on piecewise constant functions with either regular or random partitions, as well as priors defined via mixtures of Beta distributions. A comprehensive discussion of alternative approaches lies beyond the scope of this paper. A complete, though not exhaustive, comparison of Bayesian nonparametric Poisson and Hawkes models is provided in Table 3.

## 4 Neural TPPs

Benefiting from the rapid development of deep learning, another way to enhance the flexibility of TPPs is by using deep models to model TPPs. Compared to frequentist/Bayesian nonparametric TPPs, neural TPPs offer more intuitive and straightforward modeling and parameter estimation. Shchur et al. (2021) provided a comprehensive review of neural TPPs, but it focuses on work prior to 2020. This paper focuses more on the latest advancements from 2020 to 2024. In the following, we categorize neural TPPs into three major types and introduce each in detail. A schematic illustration of these models is shown in Figure 3.

Table 3: Comparison of Bayesian nonparametric Poisson and Hawkes models.

| Work | Model | Prior | Link Function | Inference | Complexity | Kernel Support |
|---|---|---|---|---|---|---|
| Adams et al. (2009) | Poisson | GP | sigmoid | MCMC | $\mathcal{O}(N^3)$ | Any |
| Gunter et al. (2014) | Poisson | Multi-output GP | sigmoid | MCMC | $\mathcal{O}(N^3)$ | Any |
| Samo & Roberts (2015) | Poisson | Sparse GP | exponential | MCMC | $\mathcal{O}(N)$ | Any |
| Flaxman et al. (2015) | Poisson | GP + Kronecker | exponential | Laplace | near $\mathcal{O}(N)$ | Any |
| Walder & Bishop (2017) | Poisson | Mercer GP | square | Laplace | $\mathcal{O}(N)$ | squared exponential |
| Sellier & Dellaportas (2023) | Poisson | Spectral GP | square | Laplace | $\mathcal{O}(N)$ | stationary |
| Lloyd et al. (2015) | Poisson | GP | square | Variational | $\mathcal{O}(N)$ | squared exponential |
| Aglietti et al. (2019b) | Poisson | Multi-output GP | exponential | Variational | $\mathcal{O}(N)$ | Any |
| Donner & Opper (2018) | Poisson | GP + Pólya-Gamma | sigmoid | Variational/MCMC/EM | $\mathcal{O}(N)$ | Any |
| Kottas & Sansó (2007) | Poisson | DP mixture | – | MCMC | not specified | – |
| Zhang et al. (2019) | Linear Hawkes | GP for $\phi(\tau)$ only | square | MCMC | $\mathcal{O}(N)$ | Any |
| Zhang et al. (2020b) | Linear Hawkes | GP for $\phi(\tau)$ only | square | Variational | $\mathcal{O}(N)$ | squared exponential |
| Zhou et al. (2021) | Linear Hawkes | GP | square | Variational | $\mathcal{O}(N)$ | squared exponential |
| Zhou et al. (2020) | Linear Hawkes | GP + Pólya-Gamma | sigmoid | Variational/MCMC/EM | $\mathcal{O}(N)$ | Any |
| Malem-Shinitski et al. (2022) | Nonlinear Hawkes | GP + Pólya-Gamma | sigmoid | Variational | $\mathcal{O}(N)$ | Any |
| Zhou et al. (2022a) | Nonlinear Hawkes | GP + Pólya-Gamma | sigmoid | Variational/MCMC/EM | $\mathcal{O}(N)$ | Any |
| Jiang & Rodriguez (2025) | Linear Hawkes | DP mixture | – | MCMC | $\mathcal{O}(N)$ | – |

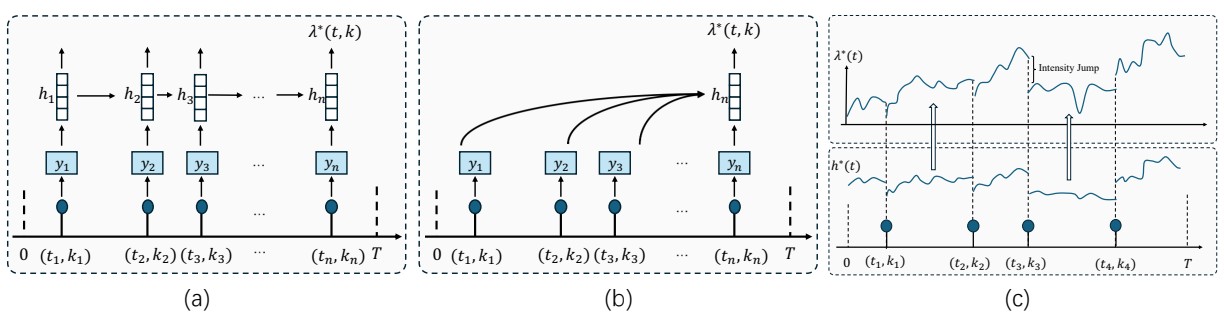

$$(a) \qquad\qquad (b) \qquad\qquad (c)$$

Figure 3: Illustration of neural TPPs. (a) Recurrent neural TPPs, where the hidden state is updated recurrently using the current event and then used to model the conditional intensity function; (b) Autoregressive neural TPPs, where the hidden state is computed by summarizing all previous events and then used to model the conditional intensity function; (c) Differential equation-based neural TPPs, where the hidden state evolves continuously when no events occur and undergoes a jump at event times, and is then used to model the conditional intensity function.

## 4.1 Recurrent Neural TPPs

The earliest work on neural TPPs can be traced back to Du et al. (2016), which was the first to use a recurrent neural network (RNN) to model TPPs. In that work, each event in $\{(t_n, k_n)\}_{n=1}^N$ is passed through an embedding layer to obtain a compact representation $\mathbf{y}_n$. At each event location, a history embedding $\mathbf{h}_n$ is designed to capture historical information. When a new event occurs, the history embedding is updated based on the feature of the current event:

$$\mathbf{h}_n = \text{Update}(\mathbf{h}_{n-1}, \mathbf{y}_n). \tag{12}$$

Finally, the history embedding $\mathbf{h}_n$ is used to parameterize the conditional distribution of the next event. In Du et al. (2016), $\mathbf{h}_n$ is used to represent the conditional intensity function after $t_n$:

$$\lambda^*(t, k) = \exp\left(\mathbf{v}_k^\top \mathbf{h}_n + w_k(t - t_n) + b_k\right), \tag{13}$$

where $\mathbf{v}_k$, $w_k$, and $b_k$ are learnable parameters for $k$-th mark. The first term captures the influence of past events through the history embedding, the second term is an extrapolation component that models the intensity at time $t$, and the third term is a bias. The exponential function ensures that the conditional intensity function remains positive. As discussed in Section 2, although the conditional intensity function is a common parameterization, other formulations for characterizing the conditional distribution of the next event are also feasible.

Due to the gradient vanishing or exploding problems of traditional RNNs, Du et al. (2016) was unable to model long-range dependencies. To address this issue, Mei & Eisner (2017) proposed a long short-term memory (Hochreiter, 1997) (LSTM)-based TPP model, which mitigates the shortcomings of traditional RNNs and achieves improved performance. Xiao et al. (2017b) introduced a dual-LSTM framework, where one LSTM models the TPP and the other models a time-series covariate. The history embeddings from both networks are then fused to predict the next event. By leveraging the additional information from covariates, TPPs can achieve improved prediction performance. Yang et al. (2018) further extended the LSTM-based point process model to the spatio-temporal domain. Omi et al. (2019) used RNNs to encode history, but modeled the cumulative conditional intensity instead of the conditional intensity itself to avoid costly numerical integration during MLE. This enables efficient training via differentiation rather than integration, as discussed in Section 4.5. Soen et al. (2021) proposed UniPoint, an RNN-based model that serves as a universal approximator for point process intensity functions. They theoretically proved that RNNs can approximate any valid intensity by leveraging the Stone-Weierstrass theorem. Gupta et al. (2021) addressed missing events by using two RNNs to model the generative processes of observed and missing events, with the latter treated as latent variables. They proposed an unsupervised training method based on variational inference to jointly learn both models. Chen et al. (2024) established excess risk bounds for RNN-based TPPs under various TPP settings. They showed that a four-layer RNN-TPP can achieve vanishing generalization error.

The main advantage of recurrent models lies in their computational efficiency. During the prediction phase, once the history embedding $\mathbf{h}_n$ is obtained, the model can predict the next event with constant time complexity $O(1)$, regardless of the sequence length. This makes recurrent models particularly suitable for online or streaming settings where fast real-time prediction is essential. Moreover, during training, the time complexity scales linearly with the number of events, i.e., $O(N)$, since the history embedding is updated in an iterative, step-by-step manner along the event sequence. However, recurrent models also suffer from several notable limitations. First, due to their inherently sequential architecture, they cannot be efficiently parallelized during training, which significantly slows down the training process, especially for long event sequences. Second, traditional RNN-based models struggle to capture long-range dependencies due to issues such as vanishing or exploding gradients, which can degrade the model's ability to learn complex temporal dynamics over extended horizons. Although architectures such as LSTMs and GRUs (Chung et al., 2014) partially mitigate these issues, they do not fundamentally resolve them. These limitations have motivated the development of alternative architectures, such as attention-based and Transformer-style models, which offer better support for parallelism and more effective modeling of long-range dependencies.

It is worth noting that in recent years, several powerful recurrent architectures have been proposed in the field of sequential models, such as the Receptance Weighted Key Value (RWKV) (Peng et al., 2023), the Structured State Space Sequence (S4) (Gu et al., 2022), and Mamba (Gu & Dao, 2023). These works aim to design efficient recurrent architectures to replace Transformers, as Transformers have a high time complexity of $O(N^2)$ in training and $O(N)$ in prediction. All these models are recurrent architectures, so they share the same advantages as RNNs, such as $O(1)$ time complexity for prediction and $O(N)$ time complexity for training. However, their key distinction from RNN lies in their ability to support parallelized training and capture long-range dependencies. Integrating these novel and efficient recurrent architectures with TPPs is an important future research direction. This integration could significantly enhance the scalability of neural TPPs for training and prediction on large-scale datasets. Currently, works in this area are limited. Gao et al. (2024) and Chang et al. (2024) explored the idea of combining Mamba or deep state space models with TPPs, offering a promising path forward for scalable TPP modeling.

## 4.2 Autoregressive Neural TPPs

As stated in Section 4.1, due to the limitations of RNNs, such as the inability to support parallel training and capture long-range dependencies, a large number of studies since 2020 have explored using Transformer architectures to model TPPs. The earliest works include Zuo et al. (2020) and Zhang et al. (2020a), which share similar ideas but differ slightly in certain details. Each event in the sequence $\{(t_n, k_n)\}_{n=1}^N$ is encoded as a feature vector $\mathbf{y}_n \in \mathbb{R}^M$, combining both temporal and mark information. We adopt sinusoidal positional

encodings for the temporal component, defined as:

$$z_j(t_n) = \begin{cases} \cos\left(t_n/10000^{\frac{j-1}{M}}\right), & \text{if } j \text{ is odd,} \\ \sin\left(t_n/10000^{\frac{j}{M}}\right), & \text{if } j \text{ is even,} \end{cases}$$

where $z_j(t_n)$ denotes the $j$-th entry of the time embedding vector $\mathbf{z}(t_n) \in \mathbb{R}^M$, and $j = 0, \ldots, M-1$. The complete time embedding matrix is denoted by:

$$\mathbf{Z} = [\mathbf{z}(t_1), \ldots, \mathbf{z}(t_N)]^\top \in \mathbb{R}^{N \times M}.$$

Let $\mathbf{U} \in \mathbb{R}^{M \times K}$ be a learnable mark embedding matrix, where $K$ is the total number of event types. The mark $k_n$ is first converted into a one-hot vector $\mathbf{k}_n \in \mathbb{R}^K$, and its embedding is given by:

$$\mathbf{e}(k_n) = \mathbf{U}\mathbf{k}_n \in \mathbb{R}^M.$$

Collecting all event mark embeddings yields:

$$\mathbf{E} = [\mathbf{e}(k_1), \ldots, \mathbf{e}(k_N)]^\top \in \mathbb{R}^{N \times M}.$$

The final embedding for each event is obtained by summing its temporal and mark embeddings:

$$\mathbf{Y} = \mathbf{Z} + \mathbf{E} \in \mathbb{R}^{N \times M},$$

where each row $\mathbf{y}_n$ in $\mathbf{Y}$ represents the complete embedding of the $n$-th event in the sequence. The matrix $\mathbf{Y}$ is then multiplied by corresponding weight matrices to compute the query, key, and value matrices: $\mathbf{Q} = \mathbf{Y}\mathbf{W}_Q \in \mathbb{R}^{N \times M_K}$, $\mathbf{K} = \mathbf{Y}\mathbf{W}_K \in \mathbb{R}^{N \times M_K}$, $\mathbf{V} = \mathbf{Y}\mathbf{W}_V \in \mathbb{R}^{N \times M_V}$. Finally, the attention score is computed as:

$$\mathbf{S} = \text{softmax}\left(\frac{\mathbf{Q}\mathbf{K}^\top}{\sqrt{M_K}}\right)\mathbf{V}. \tag{14}$$

To ensure causality and prevent future events from affecting past events, it applies a mask to the upper triangular entries of $\mathbf{Q}\mathbf{K}^\top$.

The attention score is then used to generate the history embedding $\mathbf{h}_n$, which serves as the history representation for parameterizing the conditional distribution of the next event. In Zuo et al. (2020); Zhang et al. (2020a), $\mathbf{h}_n$ is used to represent the conditional intensity function after $t_n$:

$$\lambda^*(t, k) = \text{softplus}\left(\mathbf{v}_k^\top \mathbf{h}_n + \frac{w_k(t - t_n)}{t_n} + b_k\right), \tag{15}$$

where $\mathbf{v}_k$, $w_k$, and $b_k$ are learnable parameters for $k$-th mark. However, other parameterizations for characterizing the conditional distribution of the next event are also feasible. For example, Panos (2024) used the history embedding $\mathbf{h}_n$ obtained from the Transformer to model the conditional density function of the next event $f(t \mid \mathbf{h}_n)$. This approach makes sampling the next event more efficient.

The advantages and disadvantages of autoregressive models stand in contrast to those of recurrent models. A key strength of autoregressive models lies in their ability to support parallel training and effectively capture long-range dependencies via self-attention mechanisms. However, these benefits come at the cost of computational efficiency. During training, the time and memory complexity scales quadratically with the sequence length, i.e., $\mathcal{O}(N^2)$, due to the self-attention computation across all event pairs. During prediction, autoregressive models typically require all previous hidden states to compute the next event, resulting in a time complexity of $\mathcal{O}(N)$. While key-value (KV) caching techniques can reduce the prediction time to constant time complexity $\mathcal{O}(1)$ by storing intermediate representations, they incur significant memory overhead, which can become prohibitive for long sequences or memory-constrained environments. In contrast, recurrent models maintain a hidden state that is updated incrementally, leading to linear time complexity $\mathcal{O}(N)$ for training and constant time $\mathcal{O}(1)$ for prediction, but at the cost of sequential processing and limited ability to model long-range dependencies.

Between 2020 and 2024, numerous studies have proposed improvements to Transformer TPP models. For instance, Zhu et al. (2021) modified the computation of the attention score in Equation (14), replacing the commonly used dot-product attention with a flexible non-linear attention score based on Fourier kernels, enabling the capture of more complex event similarities. Zhou et al. (2022b) extended Transformer-based point process models to the spatio-temporal domain by first using a Transformer to encode the event history. The resulting history embedding is then mapped to a latent stochastic process, which is subsequently used to generate a set of temporal and spatial kernels. These kernels are linearly combined to construct the spatio-temporal conditional intensity function. Yang et al. (2022) further improved the attention mechanism proposed in Zuo et al. (2020); Zhang et al. (2020a) by introducing future-event-specific query vectors that incorporate continuous positional encodings of time $t$. As $t$ increases, the attention weights over past events vary smoothly, enabling a more adaptive and temporally aware modeling of historical influence. Mei et al. (2021) proposed a Transformer-based model without setting the form of intensity for flexible modeling. Li & Sun (2023) proposed a sparse Transformer TPP model based on a sliding window mechanism to reduce the quadratic time and space complexity of Transformers. Meng et al. (2024) introduced improvements in the combination of time and mark embeddings, the computation of $\mathbf{Q}$ and $\mathbf{K}$ matrices, and the modeling of the conditional intensity function. These enhancements allowed the Transformer Hawkes process to perfectly align with statistical nonlinear Hawkes processes, thereby improving its interpretability. Wang et al. (2024) combined Transformer TPPs with federated learning, enabling collaborative learning from large amounts of distributed event sequence data.

### 4.3 Differential Equation-based Neural TPPs

Recurrent and autoregressive neural TPPs share a fundamental limitation: as discrete-time models, they can only compute the history embedding $\mathbf{h}_n$ and the conditional intensity function $\lambda^*(t_n)$ at discrete event times. However, they cannot directly characterize the conditional intensity function over the continuous intervals between events. To address this issue, many studies introduce extrapolation terms to approximate the intensity function over such intervals. For example, in Equation (13), the term $w_k(t - t_n)$, and in Equation (15), the term $w_k(t - t_n)/t_n$, are both designed to serve as extrapolation mechanisms. However, these extrapolation components adopt fixed parametric forms, which limit the expressiveness of the conditional intensity function over event intervals. Specifically, the extrapolation behavior in Equation (15) is approximately linear with respect to $t$, as the softplus function closely resembles a ReLU. This limitation is visually demonstrated in Meng et al. (2024), where the authors plot the conditional intensity function over intervals to highlight this issue.

Differential equation-based TPPs represent another line of research. These models, being continuous-time, can model the conditional intensity function over continuous time, thus avoiding the above issue. Specifically, these models utilize differential equations (often stochastic differential equations (SDEs)) to describe a history-dependent left-continuous hidden state $\mathbf{h}^*(t)$ over $[0, T]$ with an initial state $\mathbf{h}^*(0)$. For instance, in Jia & Benson (2019), the SDE is defined as:

$$d\mathbf{h}^*(t) = \mathrm{NN}_{\theta_1}(\mathbf{h}^*(t), t)dt + \mathrm{NN}_{\theta_2}(\mathbf{h}^*(t), t)dN(t), \tag{16}$$

where we take the unmarked case as an example. The functions $\mathrm{NN}_{\theta_1}$ and $\mathrm{NN}_{\theta_2}$ are two neural networks that govern the flow and jump of $\mathbf{h}^*(t)$, respectively. $N(t)$ is the counting process that records the number of events up to time $t$. When no event occurs, $\mathbf{h}^*(t)$ evolves smoothly according to $\mathrm{NN}_{\theta_1}$. When an event occurs, $\mathbf{h}^*(t)$ undergoes a jump at the event's timestamp, governed by $\mathrm{NN}_{\theta_2}$. Then, the history-dependent left-continuous hidden state $\mathbf{h}^*(t)$ is used to define the conditional intensity function:

$$\lambda^*(t) = \mathrm{NN}_{\theta_3}(\mathbf{h}^*(t)), \tag{17}$$

where $\mathrm{NN}_{\theta_3}$ is another neural network ensuring a non-negative output. Finally, we use MLE to estimate the model parameters. Clearly, since $\mathbf{h}^*(t)$ can vary flexibly over the event intervals—being parameterized by $\mathrm{NN}_{\theta_1}$—the resulting conditional intensity function $\lambda^*(t)$ also exhibits flexible dynamics between events. This effectively overcomes the limited expressiveness of recurrent and autoregressive neural TPPs in modeling the conditional intensity function over event intervals.

The earliest work, to the best of our knowledge, that integrates differential equations with point processes is Chen et al. (2018), which considered a simple inhomogeneous Poisson process whose intensity function evolves according to an ordinary differential equation (ODE). However, this model does not account for the discontinuities in the conditional intensity function caused by historical events. To address this limitation, Jia & Benson (2019) proposed a method based on SDE, as shown in Equations (16) and (17), to model the conditional intensity of history-dependent TPPs. This approach allows the conditional intensity function to exhibit jumps at the occurrence of events. A contemporaneous work, Rubanova et al. (2019), combined ODEs with RNNs. In this framework, the hidden state $\mathbf{h}^*(t)$ evolves smoothly according to an ODE when no event occurs, and undergoes a jump at the event time, governed by the RNN update. This enables the model to capture jumps in the conditional intensity function at event times. In fact, Rubanova et al. (2019) shares strong similarities with Mei & Eisner (2017), but is more general. Specifically, Mei & Eisner (2017) assumes that $\mathbf{h}^*(t)$ evolves across event intervals following an exponential law, whereas Rubanova et al. (2019) allows $\mathbf{h}^*(t)$ to evolve flexibly over event intervals according to arbitrary ODEs. The ODE-based approach was later extended to spatio-temporal TPPs by Chen et al. (2021). Wang et al. (2018) adopted a SDE-based approach to directly model the conditional intensity function of TPPs. Additionally, they used an SDE to model users' opinions, enabling the incorporation of user feedback into the TPP framework. This results in a closed-loop model that jointly captures the dynamics of users' opinions and event generation. More recently, Zhang et al. (2024) proposed a novel SDE-based method for modeling TPPs. Instead of modeling a latent hidden state via an SDE and mapping it to the conditional intensity function, their approach directly models the conditional intensity function with an SDE, and further provides a theoretical analysis on the existence and uniqueness of its solution.

Although differential equation-based neural TPPs offer greater flexibility than recurrent and autoregressive models in characterizing the conditional intensity function over event intervals, they suffer from significant computational inefficiencies during both training and sampling. During training, when using MLE, the model requires numerical integration to compute the integral of the conditional intensity function. This necessitates evaluating the intensity at multiple time points, which can only be obtained by solving the ODE or SDE defined in Equation (16) using a numerical solver. As a result, training is considerably slower compared to recurrent or autoregressive neural TPPs. Similarly, during the sampling phase—such as when applying the thinning algorithm—intensity values at various time points must also be computed via ODE or SDE solvers, further increasing the computational cost and slowing down the sampling process. A comparison among the three mainstream neural TPP architectures is provided in Table 4.

Table 4: Comparison of neural TPP architectures.

| Model Type | Parallel Training | Long-range | Training Complexity | Prediction Complexity | Continuous-time | Strengths | Limitations |
|---|---|---|---|---|---|---|---|
| Recurrent | ✗ | Limited | $O(N)$ | $O(1)$ | No | efficient prediction | slow training |
| Transformer | ✔ | Strong | $O(N^2)$ | $O(N)/O(1)^4$ | No | parallel training | high complexity |
| ODE/SDE-based | ✗ | Strong | depend on solver | depend on solver | Yes | continuous-time, expressive | slow training/prediction |

## 4.4 Diffusion-based TPPs

Recently, diffusion-based generative modeling has emerged as a promising new direction for TPPs. Diffusion models have achieved remarkable success in a wide range of generative tasks, including image generation, video synthesis, and tabular data modeling. In these models, data are generated by reversing a gradual noising process through iterative denoising steps (Song et al., 2020).

Inspired by these advances, diffusion models have recently been introduced into temporal point process modeling. Unlike conventional neural TPPs that typically model the conditional intensity function or inter-event distribution in an autoregressive manner, diffusion-based approaches aim to learn the distribution of an entire event sequence through iterative denoising. This perspective enables non-autoregressive sequence generation and can mitigate the error accumulation problem that often arises in long-horizon prediction.

One of the first works in this direction is Lüdke et al. (2023), which develops a diffusion framework specifically tailored to temporal point processes. Instead of applying standard Gaussian diffusion directly, the

---

[4]with KV cache.

method constructs a noising and denoising process through point addition and thinning operations, which are classical mechanisms in point-process simulation. The reverse diffusion process therefore naturally preserves the semantics of event sequences while learning the underlying intensity structure. Subsequent work has further extended diffusion modeling for event sequences. For instance, Kerrigan et al. (2024) proposes a diffusion-based approach that predicts multiple future events within a prediction horizon through a single denoising trajectory, rather than repeatedly predicting the next event in an autoregressive fashion. This design allows the model to capture global dependencies among future events. Other studies have explored diffusion modeling for more complex point-process structures. For example, Yuan et al. (2023) introduce a diffusion-based framework for jointly modeling spatial and temporal point processes, enabling unified generation of spatiotemporal event patterns while addressing the conditional dependence between spatial and temporal dynamics. Lüdke et al. (2024) formulate point processes as unordered sets and apply point-cloud denoising techniques to enable permutation-invariant event generation and direct likelihood evaluation.

Compared with traditional intensity-based or autoregressive neural TPPs, diffusion-based approaches offer a complementary modeling paradigm. By generating entire event sequences through iterative denoising, they provide a more global view of future event trajectories and are potentially better suited for long-horizon forecasting and sequence simulation. However, diffusion-based (non-autoregressive) generation also introduces several important limitations. First, unlike autoregressive models that explicitly model the conditional distribution of the next event given history, diffusion models learn a global sequence distribution through iterative denoising. This makes it less straightforward to enforce temporal consistency and causal dependence across events, especially for long sequences. Second, the multi-step denoising process significantly increases both training and inference cost, as generating a single sequence typically requires dozens or even hundreds of refinement steps. Third, non-autoregressive generation lacks an explicit likelihood formulation in many cases, which makes model evaluation, calibration, and comparison with classical TPP methods more difficult. Finally, diffusion models may struggle to accurately capture fine-grained temporal structures (e.g., inter-event time distributions or intensity dynamics), since temporal information is implicitly represented in the denoising trajectory rather than directly parameterized. These limitations highlight a fundamental trade-off between global sequence modeling and efficient, interpretable conditional prediction in TPPs.

### 4.5 Parametrization of Neural TPPs

Both recurrent and autoregressive neural TPPs extract a history embedding from past event information and use it to model the conditional distribution of the next event. As discussed in Section 2, there are multiple ways to represent the conditional distribution of the next event, such as the conditional density function $f(t \mid \mathcal{H}_{t_n})$ in Equation (1), the cumulative distribution function $F(t \mid \mathcal{H}_{t_n})$ in Equation (2), the conditional intensity function $\lambda^*(t)$ in Equation (2), and the cumulative intensity function $\Lambda^*(t) = \int_0^t \lambda^*(\tau)d\tau$. All of these parameterizations are equivalent and can be used to characterize the conditional distribution of the next event, since, as proved in Equation (3), the conditional intensity function and the conditional density function are in a one-to-one correspondence. A schematic illustration of the four common parameterizations of TPPs is shown in Figure 4.

Most of the works discussed above adopt the conditional intensity function as the primary parameterization. The main advantage of this approach lies in its conceptual and implementation simplicity—the model only needs to ensure that the output is non-negative, which can be easily achieved using functions like ReLU, softplus, or exponential mappings. However, a notable drawback arises during MLE: the log-likelihood requires evaluating the integral of the conditional intensity function over the observation window. In most practical cases, this integral does not admit a closed-form solution and must be approximated using numerical integration techniques, such as Monte Carlo sampling or quadrature. This introduces additional computational overhead and can compromise both the estimation accuracy and the overall training efficiency, especially for complex or high-dimensional models.

In recent years, several studies have explored alternative parameterizations for TPPs. Shchur et al. (2020a) and Panos (2024) modeled the conditional density function $f(t \mid \mathcal{H}_{t_n})$ directly as a mixture of log-normal distributions, using history embeddings as inputs. The key difference between the two lies in how the history embeddings are extracted: Shchur et al. (2020a) used a RNN, while Panos (2024) employed a Transformer-

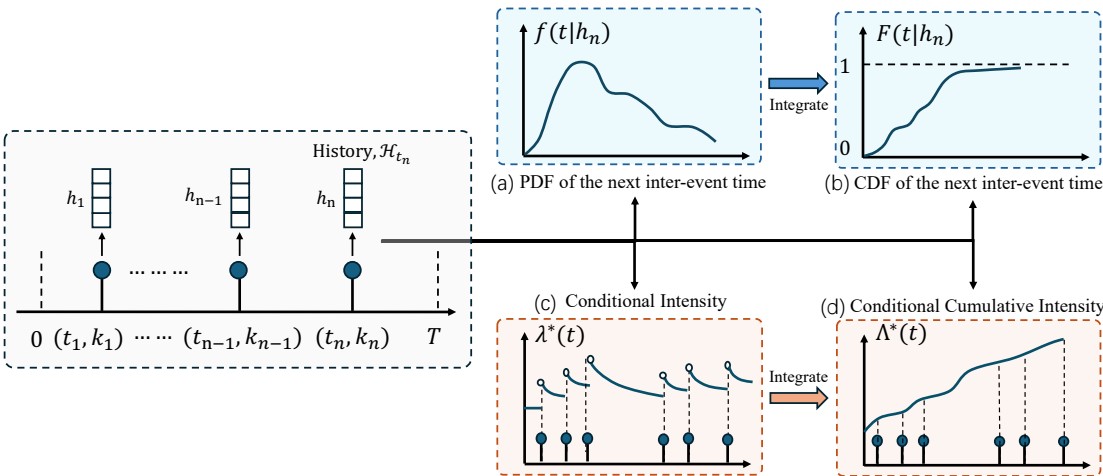

Figure 4: Four common parameterizations of TPPs: (a) the probability density function (PDF) of the next event, (b) the cumulative distribution function (CDF) of the next event, (c) the conditional intensity function, and (d) the conditional cumulative intensity function.

based encoder. One major advantage of modeling the conditional density directly is that it eliminates the need to compute the integral of the conditional intensity function during MLE, earning it the name "intensity-free" modeling. Another benefit is efficient sampling: given the history, the next event can be sampled directly from a mixture of log-normal distributions, which admits a simple and tractable form. Taieb (2022) modeled the inverse of the cumulative distribution function $F^{-1}(t \mid \mathcal{H}_{t_n})$ using the history embedding with monotonic rational-quadratic splines. This method also avoids numerical integration and supports efficient sampling. Omi et al. (2019); Shchur et al. (2020b); Liu (2024) proposed modeling the cumulative intensity function $\Lambda^*(t)$ conditioned on history using monotonic neural networks or splines. This enables a reformulation of the log-likelihood in Equation (7), which, for the unmarked case, becomes:

$$\log f(\mathcal{T}; \theta) = \sum_{n=1}^{N} \log \frac{d}{dt} \Lambda_\theta^*(t_n^-) - \Lambda_\theta^*(T),\tag{18}$$

where $t_n^-$ denotes the left-hand limit of the derivative at $t_n$. By modeling the cumulative intensity function directly, this parameterization transforms the integral in the log-likelihood into a derivative, allowing for efficient computation using automatic differentiation. Consequently, it eliminates the need for numerical integration during MLE training, improving both accuracy and efficiency. The comparison of four parameterizations of TPPs is provided in Table 5.

Table 5: Comparison of neural TPP parameterizations.

| Parameterization | Numerical Integration | Sampling Efficiency | Training Efficiency | Requirement | Representative Works |
|---|---|---|---|---|---|
| Conditional Intensity Function $\lambda^*(t \mid \mathcal{H})$ | Yes | Low | Low | Nonnegative | (Du et al., 2016; Mei & Eisner, 2017; Zuo et al., 2020) |
| Conditional Density Function $f(t \mid \mathcal{H})$ | No | High | High | Nonnegative, normalized | (Shchur et al., 2020a; Panos, 2024) |
| Cumulative Distribution Function $F(t \mid \mathcal{H})$ | No | High | High | Monotonically increasing between $(0,1)$ | (Taieb, 2022) |
| Cumulative Intensity Function $\Lambda^*(t \mid \mathcal{H})$ | No | Low | High | Nonnegative, monotonically increasing | (Omi et al., 2019; Shchur et al., 2020b; Liu, 2024) |

# 5   LLM-based TPPs

LLM-based temporal point processes can be broadly categorized into two paradigms depending on the role played by large language models. One line of work uses LLM-inspired mechanisms to enhance conventional neural TPPs without replacing their temporal modeling backbone, while another line directly integrates LLMs as the core sequence model for representing and predicting event streams. Although both aim to leverage the representational power of LLMs, they differ fundamentally in architecture, training strategy, and how semantic and temporal information are encoded. More broadly, this line of research lies at the

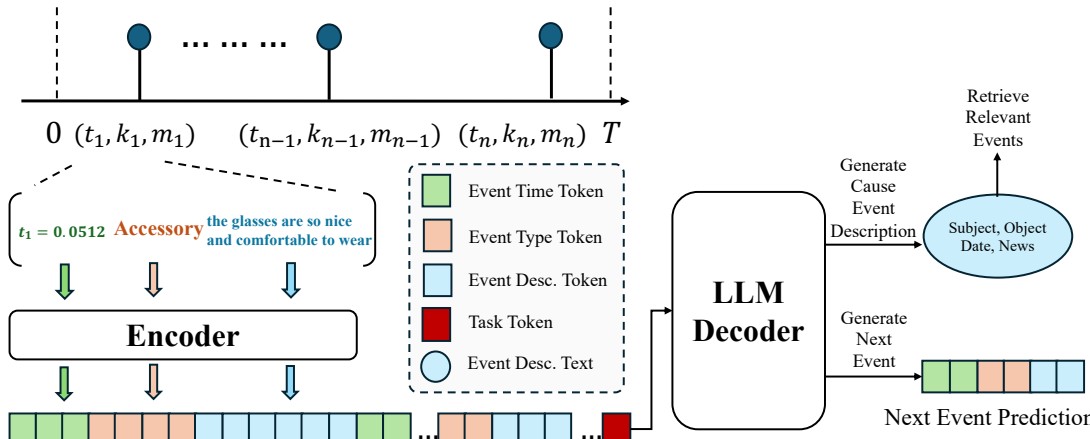

Figure 5: An overview of LLM-based TPPs. The event time $t$, type $k$, and associated multimodal data $m$ are first encoded into tokens via an encoder. These tokens are then fed into a LLM, which can be used to generate the next event or produce a textual description of the causal events. The encoder design varies across different works. For example, Liu & Quan (2024) uses temporal positional encodings as in standard Transformers; Shi et al. (2024) treats event times as text and leverages the LLM's built-in tokenizer; and Kong et al. (2025) converts event times into floating numbers and encodes them as byte-level tokens.

intersection of temporal point processes, sequential foundation models, multimodal event understanding, and retrieval-augmented reasoning. Unlike classical TPPs, which mainly focus on continuous-time likelihood modeling and prediction, LLM-based approaches naturally connect event streams with free text, external knowledge, and heterogeneous modalities. This makes them relevant not only to TPP modeling, but also to adjacent areas such as temporal representation learning, event-sequence retrieval, multimodal reasoning, and question answering over time-stamped observations. From this perspective, LLM-based TPPs should be understood not merely as a new model family, but as an expansion of the TPP research agenda toward semantically rich event understanding. A schematic illustration of LLM-based TPPs is shown in Figure 5.

## 5.1 LLM-inspired TPPs

LLM-inspired TPPs retain a neural temporal point process as the primary model of event dynamics but borrow ideas from LLMs—such as prompt learning or reasoning—to improve adaptation and interpretability. PromptTPP (Xue et al., 2023b) is a representative example that introduces prompt learning into neural TPPs to enable continual learning under distribution shifts. Instead of retraining the base TPP when new data arrive, PromptTPP prepends a small set of learnable temporal prompts to the event sequence. These prompts are retrieved from a continuously updated memory pool and optimized jointly with the TPP, allowing the model to adapt to new patterns without storing past data or introducing task-specific modules. This design is computationally efficient and particularly attractive in streaming or privacy-constrained scenarios. However, because the prompts only modulate an existing neural TPP, the approach does not introduce higher-level semantic or causal reasoning.

A different LLM-inspired strategy is adopted in LAMP (Shi et al., 2024), which augments neural TPPs with abductive reasoning capabilities provided by an LLM. LAMP uses a multi-stage pipeline in which a base event sequence model first proposes candidate future events. A fine-tuned LLM then generates plausible causes for each proposed event, after which a retrieval module searches the historical sequence for events matching these causes. A scoring function evaluates whether the retrieved events can reasonably explain the proposed future events. In contrast to PromptTPP, which focuses on efficient adaptation via prompts, LAMP aims to improve prediction quality and interpretability through causal-style reasoning. However, this comes at the cost of increased complexity and reduced robustness, since prediction depends on multiple interacting components, while the temporal dynamics themselves are still modeled by a conventional TPP.

## 5.2 Direct LLM-TPP Integration

In contrast, direct LLM–TPP integration methods use LLMs as the primary model for representing and predicting event sequences, embedding both semantic and temporal information into the LLM input space. TPP-LLM (Liu & Quan, 2024) follows this paradigm by representing events through their textual descriptions rather than categorical marks, enabling pretrained LLMs to capture rich semantic relations between events. Temporal information is injected via positional-style temporal embeddings, and parameter-efficient fine-tuning methods such as LoRA are employed to adapt the LLM to temporal prediction tasks. The LLM produces contextualized representations of the event history, which are then passed to an intensity head for predicting the next event time and type. Compared to LLM-inspired approaches, TPP-LLM tightly couples semantic understanding with temporal modeling, but it still relies on external temporal embeddings to represent continuous time.

Language-TPP (Kong et al., 2025) proposes a more unified representation by encoding continuous time intervals directly as byte-level tokens that are processed by the LLM in the same way as natural language. This eliminates the need for positional or temporal embeddings and allows the LLM to model time and semantics in a single token sequence. As a result, Language-TPP supports standard TPP tasks such as event time and type prediction, while also enabling new capabilities such as generating natural-language event descriptions. Compared with TPP-LLM, this token-based temporal encoding provides a tighter integration of time and language but leads to longer sequences and higher computational cost.

Overall, LLM-inspired TPPs such as PromptTPP and LAMP enhance existing neural TPPs with adaptation and reasoning mechanisms, whereas direct LLM–TPP approaches such as TPP-LLM and Language-TPP redefine TPP modeling by using LLMs as the core event sequence model. This distinction highlights different trade-offs in efficiency, expressiveness, interpretability, and scalability across the emerging landscape of LLM-based temporal point processes.

## 5.3 Other Extensions

Liu & Quan (2025) introduced TPP-Embedding for temporal event sequence retrieval from textual descriptions, targeting applications in e-commerce behavior analysis, social media monitoring, and criminal incident tracking. The authors developed TESRBench, a comprehensive benchmark with diverse real-world datasets and synthesized textual descriptions. Their model leverages the TPP-LLM (Liu & Quan, 2024) framework to integrate LLMs with TPPs, encoding both event texts and temporal information through pooling representations and contrastive loss to align sequence-level embeddings with textual descriptions. This approach outperforms baseline models across TESRBench datasets and establishes foundations for retrieval-augmented generation in TPP domains.

Extending beyond textual data, Jiang et al. (2025) introduced DanmakuTPPBench, a comprehensive multi-modal TPP benchmark addressing the gap in datasets with temporal, textual, and visual information. The benchmark comprises two components: DanmakuTPP-Events, derived from Bilibili's user-generated bullet comments (Danmaku) that form multi-modal events with timestamps, textual content, and video frames; and DanmakuTPP-QA, a question-answering dataset constructed via a multi-agent pipeline using state-of-the-art LLMs and multi-modal LLMs (MLLMs). Targeting temporal-textual-visual reasoning tasks requiring multi-modal event dynamics understanding, extensive evaluations of classical TPP models and recent MLLMs reveal significant performance gaps in modeling multi-modal event sequences. This work establishes baselines and opens research directions for integrating TPP modeling into the multi-modal language modeling landscape. An illustration of DanmakuTPPBench is shown in Figure 6.

**On the boundary of the TPP framework.** Recent LLM-based extensions significantly broaden the scope of TPP research by introducing tasks such as event-sequence retrieval, question answering, and multimodal reasoning. While these tasks are highly valuable, they do not always fit neatly into the traditional definition of a temporal point process as a stochastic process over event times and marks. In classical TPPs, the central object is a probability law over event occurrences in continuous time, and core tasks focus on likelihood modeling, prediction, simulation, or causal structure discovery. By contrast, retrieval or QA tasks

often require high-level semantic understanding, external knowledge integration, and cross-modal reasoning that go beyond the standard probabilistic objectives of TPPs.

This observation does not diminish the importance of these new directions; rather, it highlights a conceptual shift. LLM-based TPP research is gradually moving from modeling event occurrence processes to understanding temporally indexed event data. As a result, future work may need to distinguish more clearly between tasks that are fundamentally point-process problems and tasks that use TPPs as one component within a broader temporal reasoning system. Clarifying this boundary would help the community better define evaluation protocols, modeling assumptions, and the scope of claimed contributions.

Overall, this direction is rapidly evolving, with approaches ranging from LLM-inspired techniques to direct integration and multi-modal extensions. The field shows promise for exploration in areas such as retrieval-augmented TPPs, multi-modal event understanding, and more sophisticated temporal reasoning capabilities.

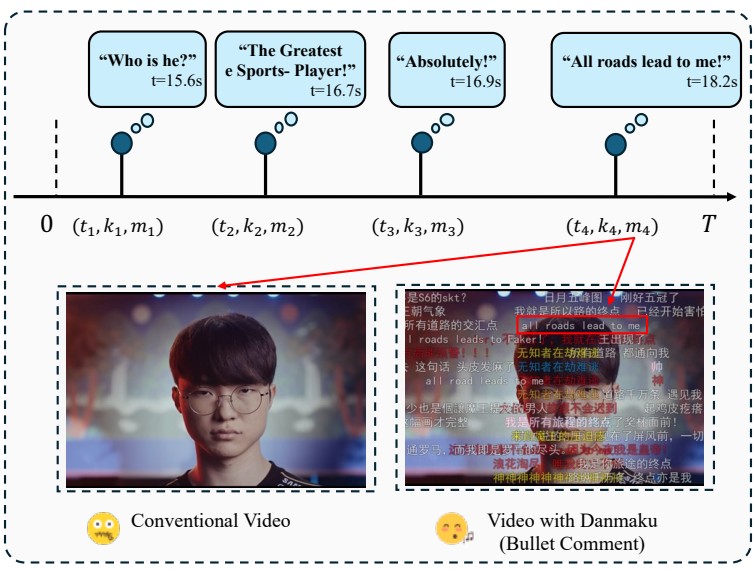

Figure 6: An illustration of DanmakuTPPBench (Jiang et al., 2025), a multi-modal benchmark for TPPs. The dataset consists of time-stamped events with associated event types $k$ and multimodal information $m$, where $m$ includes both the textual content of user comments (danmaku) and the corresponding video frames. This benchmark establishes standard baselines and opens new research directions for integrating TPP modeling into the broader landscape of multimodal language modeling.

## 6 Datasets, Benchmarks, and Evaluation Protocols

Besides model design, progress in TPPs also depends critically on datasets, benchmark protocols, and evaluation metrics. In practice, however, the empirical study of TPPs has long suffered from fragmented datasets, inconsistent preprocessing, different train/validation/test splits, and heterogeneous metric definitions. As a result, performance comparisons across papers are often not directly comparable. Recent efforts such as EasyTPP (Xue et al., 2023a) have started to address this issue by providing a unified benchmarking framework, standardized implementations, and common evaluation pipelines. In this section, we summarize representative datasets, typical evaluation tasks, and commonly used metrics.

**Representative datasets.** Existing TPP datasets cover a wide range of domains, including social interactions, e-commerce, finance, healthcare, and multimodal user behavior. Classical datasets frequently used in neural TPP studies include Retweet (Zhou et al., 2013a), StackOverflow (Du et al., 2016), and Taxi (Whong, 2014), which are typically medium-scale and mostly focus on timestamp–type prediction. More recently, larger-scale datasets have become available. For example, Amazon review data (Ni et al., 2019) provide large collections of user-item interactions with timestamps and rich semantic information, making

them suitable for studying large-scale event modeling and language-enhanced TPPs. Taobao-based datasets (Alibaba, 2018) further provide realistic industrial event streams with dense user behaviors and are particularly useful for evaluating long-horizon prediction and large-scale sequential decision settings. In addition, benchmark-oriented resources such as DanmakuTPPBench (Jiang et al., 2025) extend the scope of TPP evaluation to text retrieval, question answering, and multimodal temporal reasoning, broadening the traditional view of event sequence modeling.

**Benchmark standardization.** A major recent development is the emergence of unified benchmark toolkits. EasyTPP is one of the first systematic efforts toward open benchmarking for TPPs, providing standardized data preprocessing, model implementations, training pipelines, and evaluation scripts (Xue et al., 2023a). Such a benchmark is valuable not only for fair comparison, but also for improving reproducibility and lowering the entry barrier for researchers new to the field. From the perspective of a survey paper, benchmark standardization is as important as new model classes, because it determines whether empirical conclusions can be trusted and accumulated across studies.

**Evaluation tasks.** Existing TPP evaluation protocols can be roughly divided into four categories. The first is *next-event prediction*, where the goal is to predict the time and/or mark of the next event. This is the most common setting in the literature. The second is *long-horizon prediction*, where the model predicts multiple future events over an extended time window rather than only the next one. This task is substantially more difficult because errors accumulate autoregressively and the model must capture longer-term uncertainty. HyPro (Xue et al., 2022) explicitly highlights this setting and proposes a benchmark protocol for evaluating long-horizon event-sequence prediction. The third category is *semantic or multimodal tasks*, such as sequence retrieval, question answering, and multimodal reasoning, which arise in recent LLM-based TPP benchmarks such as DanmakuTPPBench (Jiang et al., 2025).

**Evaluation metrics.** The choice of metrics depends on the task. For next-event prediction, common metrics include time prediction error (e.g., MAE or RMSE of the next-event time), and classification metrics such as accuracy or macro-F1 for mark prediction. For long-horizon prediction, one often needs sequence-level metrics that assess the quality of multi-step forecasts over a future window, rather than only one-step accuracy. In this case, both temporal and mark-distribution alignment become important. One may consider distributional metrics such as negative log-likelihood, Wasserstein-style discrepancies or statistics derived from inter-event times and event-type frequencies. For retrieval and QA benchmarks in LLM-based TPPs, information-retrieval metrics and task-specific accuracy measures are also needed. Therefore, a complete evaluation of TPP models should align the metric with the intended downstream use, rather than relying only on likelihood.

**Empirical Insights.** Although existing benchmarks provide diverse evaluation settings, several consistent empirical patterns can be observed across prior studies. First, neural TPPs, especially Transformer-based models, generally outperform classical parametric models (e.g., Hawkes processes) in next-event prediction tasks, particularly on large-scale and complex datasets. Second, models that directly parameterize the conditional density or cumulative intensity function often achieve better training efficiency and comparable or superior predictive accuracy compared to intensity-based models, due to the avoidance of numerical integration. Third, long-horizon prediction remains challenging for all model classes, with autoregressive methods suffering from error accumulation, while non-autoregressive approaches (e.g., diffusion-based models) may struggle to maintain temporal consistency. Finally, recent LLM-based and multimodal TPP approaches demonstrate improved performance in tasks involving semantic understanding or heterogeneous data, but their advantages are less clear in purely temporal prediction benchmarks. These observations suggest that model performance is highly task-dependent, and no single modeling paradigm consistently dominates across all evaluation settings.

## 7 Model Training

In this section, we focus on frequentist methods for estimating model parameters in neural TPPs and LLM-based TPPs. Let $\mathcal{T} = \{(t_n, k_n)\}_{n=1}^{N}$ denote an observed event sequence and let $f_\theta(\mathcal{T})$ be the model

distribution. Parameter learning is commonly formulated as minimizing a discrepancy between the empirical data distribution and the model distribution:

$$\hat{\theta} = \arg\min_{\theta} \mathrm{D}(f(\mathcal{T}) \,\|\, f_{\theta}(\mathcal{T})). \tag{19}$$

Depending on the choice of D, one obtains different estimators with distinct statistical and computational trade-offs. To make these differences explicit, we next summarize the objective functions and practical properties of four representative training principles.

## 7.1 KL Divergence

KL divergence is a commonly used training criterion. Minimizing the KL divergence $\mathrm{KL}(f(\mathcal{T}) \,\|\, f_{\theta}(\mathcal{T}))$ is equivalent to maximizing the log-likelihood:

$$\hat{\theta} = \arg\max_{\theta} \log f_{\theta}(\mathcal{T}).$$

For a marked TPP with conditional intensity $\lambda_{\theta}^{*}(t, k)$, the log-likelihood of an observed sequence over $[0, T]$ can be written as

$$\log f_{\theta}(\mathcal{T}) = \sum_{n=1}^{N} \log \lambda_{\theta}^{*}(t_n, k_n) - \int_{0}^{T} \sum_{k=1}^{K} \lambda_{\theta}^{*}(\tau, k) \, d\tau. \tag{20}$$

Therefore, minimizing the KL divergence is equivalent to maximizing equation 20. The first term rewards high intensity at observed events, while the second term normalizes the process by penalizing excessive intensity mass over the observation window.

This approach necessitates explicit evaluation of the conditional intensity function and its integral over time, which is typically intractable and must be approximated via numerical integration. Despite its computational burden, MLE remains asymptotically efficient and statistically optimal. Due to its solid theoretical foundation and general applicability, MLE has been adopted in the vast majority of TPP studies (Ozaki, 1979; Paninski, 2004).

## 7.2 Wasserstein Distance

In Wasserstein-based training, the model is learned by minimizing a distributional distance between model and data distributions:

$$\hat{\theta} = \arg\min_{\theta} W\big(f(\mathcal{T}), f_{\theta}(\mathcal{T})\big), \tag{21}$$

where $W(\cdot, \cdot)$ denotes the Wasserstein distance. Since the Wasserstein distance is generally difficult to compute directly, it is typically approximated using an adversarial framework with a critic network based on the Kantorovich–Rubinstein dual formulation. For example, Xiao et al. (2017a) proposed leveraging the Wasserstein distance within a Wasserstein GAN (WGAN) framework for TPPs. In this setup, the TPP model acts as a generator, and a critic network learns to distinguish between real and generated event sequences by minimizing the Wasserstein distance.

Compared with KL divergence, the Wasserstein distance avoids the numerical integration required in MLE, yields smoother gradients for optimization, and is generally more robust to mode collapse. However, the adversarial training procedure used in WGANs may introduce additional instability and increase training complexity. To further improve upon this approach, Xiao et al. (2018) introduced a likelihood-free training method based directly on the Wasserstein distance between point processes. Unlike the earlier WGAN-based framework, which primarily learns the aggregate intensity over a dataset, their method enables individual-level, in-sample forward prediction of event sequences conditioned on historical context. Moreover, Wasserstein distance captures the underlying geometric structure of event sequences more effectively than KL divergence, leading to more robust alignment between the generated and real sequences.

### 7.3 Noise Contrastive Estimation

Noise contrastive estimation (NCE) reframes parameter learning as a binary classification problem that distinguishes observed data from artificially generated noise samples, thereby bypassing direct likelihood computation and making it suitable for models with intractable likelihoods such as TPPs. A generic objective takes the form

$$\hat{\theta} = \arg\max_{\theta} \mathbb{E}_{\mathcal{T} \sim f}\left[ \log \sigma(s_\theta(\mathcal{T})) \right] + \mathbb{E}_{\widetilde{\mathcal{T}} \sim q}\left[ \log(1 - \sigma(s_\theta(\widetilde{\mathcal{T}}))) \right], \tag{22}$$

where $f$ denotes the observed TPP, $q$ a noise TPP, $s_\theta(\cdot)$ a score function, and $\sigma(\cdot)$ the sigmoid function.

Both Guo et al. (2018) and Mei et al. (2020) applied NCE techniques to estimate the parameters of TPPs. The key difference between them lies in the specific NCE variant they adopt. Guo et al. (2018) employed the original binary classification formulation proposed by Gutmann & Hyvärinen (2012), which treats the learning task as discriminating between real and noise sequences. In contrast, Mei et al. (2020) adopted a ranking-based NCE variant introduced by Jozefowicz et al. (2016), which is better suited for modeling conditional distributions—a natural fit for TPPs where future events are conditioned on historical ones.

### 7.4 Fisher Divergence

Fisher divergence, also known as score matching (Hyvärinen, 2005), measures discrepancies between distributions through their score functions (i.e., gradients of log-densities). Since it does not require evaluating normalizing constants, it is particularly attractive for models with intractable likelihoods. In general form, the objective can be written as

$$\hat{\theta} = \arg\min_{\theta} \mathbb{E}_f \left[ \|\nabla \log f(\mathcal{T}) - \nabla \log f_\theta(\mathcal{T})\|^2 \right], \tag{23}$$

although defining the score operator for TPPs is more subtle than in the standard Euclidean-density setting.

Several prior works have introduced score matching into the context of point processes. For example, Sahani et al. (2016) derived score matching estimators for classical Poisson processes. Zhang et al. (2023) extended this framework to deep covariate-based spatio-TPPs, while Li et al. (2023) further generalized it to multivariate Hawkes processes. These efforts have significantly advanced the applicability of score matching in point process modeling. However, recent work by Cao et al. (2024; 2025) highlights a critical limitation: the estimators proposed in these earlier studies are incomplete and only valid for specific classes of point processes. In more general cases—including some simple parametric models—these methods fail to produce accurate parameter estimates. To address this issue, Cao et al. (2024; 2025) proposed a weighted (autoregressive) score matching estimator that generalizes to a broader class of point process models, offering improved theoretical soundness and practical applicability.

### 7.5 Comparison between Different Estimators

Among these methods, only MLE requires computing the integral of the conditional intensity function. The others avoid this step, potentially improving training efficiency. All methods are consistent under mild conditions, but MLE remains asymptotically optimal in terms of variance. Alternative methods may exhibit slower convergence and higher asymptotic variance, but offer better scalability and computational simplicity. A systematic comparison of representative training objectives for TPPs is summarized in Table 6, highlighting their computational requirements, statistical properties, and practical trade-offs.

Table 6: Comparison of TPP training objectives.

| Estimator | Divergence Type | Likelihood Required | Numerical Integration | Statistical Efficiency | Representative Works |
|---|---|---|---|---|---|
| MLE | KL divergence | Yes | Yes | Asymptotically optimal | (Ozaki, 1979; Paninski, 2004) |
| Wasserstein | Wasserstein distance | No | No | Consistent, higher variance | (Xiao et al., 2017a; 2018) |
| NCE | Classification-based | No | No | Consistent, higher variance | (Guo et al., 2018; Mei et al., 2020) |
| Score Matching | Fisher divergence | No | No | Consistent, higher variance | (Sahani et al., 2016; Li et al., 2023; Cao et al., 2024) |

Table 7: Comparison of TPP applications in event prediction and causal discovery.

| Application Type | Domain | Primary Objective | Event Representation | Typical Models | Representative Works |
|---|---|---|---|---|---|
| Event Prediction | Social Networks | Predict future event time/type | User actions (posts, retweets) | Hawkes, Neural TPPs | (Kobayashi & Lambiotte, 2016; Zhang et al., 2021) |
| Event Prediction | Epidemiology | Forecast disease spread | Infection times, locations | Hawkes, Spatio-temporal TPPs | (Rizoiu et al., 2018; Chiang et al., 2022) |
| Event Prediction | Earthquakes | Aftershock forecasting | Time–location of quakes | Spatio-temporal Hawkes | (Ogata, 1998; Kwon et al., 2023) |
| Event Prediction | Finance | Market event prediction | Trades, orders | Hawkes, Neural Hawkes | (Bacry & Muzy, 2014; Shi & Cartlidge, 2022) |
| Event Prediction | Recommendation | User behavior prediction | Purchases, clicks | Neural TPPs | (Mei & Eisner, 2017; Wang et al., 2021) |
| Causal Discovery | Neuroscience | Infer functional connectivity | Neural spike trains | Multivariate Hawkes | (Linderman & Adams, 2015; Zhou et al., 2022a) |
| Causal Discovery | Finance | Discover buy–sell influence | Order book events | Hawkes with sparsity | (Bacry & Muzy, 2014; Xu et al., 2016) |
| Causal Discovery | AI Operations | Root cause analysis | System failure events | Topological Hawkes | (Cai et al., 2022) |
| Causal Discovery | Healthcare | Analyze treatment interactions | Symptoms, drug events | Hawkes-based causal models | (Bao et al., 2017) |
| Causal Discovery | Cybersecurity | Attack pattern analysis | Security alerts | Neural Hawkes | (Fortino et al., 2022) |

# 8 Applications

TPPs have a wide range of applications, including in seismology, finance, neuroscience, social networks, and epidemiology. Broadly speaking, these applications can be categorized into two main types: event prediction and causal discovery. To provide a structured overview, Table 7 summarizes representative TPP applications across different domains, highlighting their objectives, modeling choices, and typical use cases.

## 8.1 Application in Event Prediction

Event prediction leverages historical data to forecast the timing, frequency, and types of future events, with applications spanning social networks, epidemiology, earthquake forecasting, finance, and recommendation systems. In social networks, Hawkes processes and neural TPPs are widely employed to model temporal interactions and information diffusion. For instance, Zhang et al. (2021) introduced a neural TPP for detecting coordinated behavior, while Kobayashi & Lambiotte (2016); Hegde et al. (2022) applied Hawkes processes to retweet dynamics prediction, and Cencetti et al. (2021) further analyze the higher-order social interactions. Recent work enhances interpretability (Meng et al., 2024) and predicting information popularity (Li et al., 2025). Additionally, these methods aid in anomaly detection, such as identifying fake accounts (Qu et al., 2022), and Ahammad (2024) integrated sentiment analysis to detect fake news trends during the COVID-19 pandemic.

In epidemiology, Hawkes processes effectively model disease propagation, as demonstrated by Rizoiu et al. (2018). Studies such as Chiang et al. (2022) incorporated mobility data to predict COVID-19 transmission patterns, while Schwabe et al. (2021) leveraged similar data for early outbreak forecasting. Further contributions include estimating transmission times (Schoenberg, 2023) and assessing pandemic impacts using self-exciting processes (Giudici et al., 2023). In earthquake forecasting, spatio-temporal Hawkes processes are instrumental in capturing aftershock sequences. For instance, Ogata (1998) introduced spatio-temporal TPP models for earthquake occurrences, with recent advancements enhancing flexibility (Kwon et al., 2023) and refining decay rate modeling (Davis et al., 2024). In financial markets, Hawkes processes analyze market microstructure and limit order book dynamics (Chen et al., 2022a). Neural Hawkes processes (Shi & Cartlidge, 2022) and (Nyström & Zhang, 2022) further improve predictive accuracy in high frequency financial data. Recommendation systems leverage Hawkes processes to model sequential user behavior. Wang et al. (2021) combined them with attention mechanisms for sequential recommendations. These techniques can also be used to predict a user's future shopping times and item types based on their past purchase history, enabling targeted promotional strategies (Mei & Eisner, 2017; Meng et al., 2024).

Beyond event prediction, TPPs can also be used to analyze heterogeneous event streams and uncover latent temporal patterns. One representative task is event sequence clustering, which aims to group event sequences according to their underlying temporal dynamics. This is useful in applications such as user behavior analysis, patient stratification, and social activity mining, where different groups may exhibit distinct triggering patterns or interaction structures. A representative approach is the Dirichlet mixture model of Hawkes processes proposed by Xu & Zha (2017), which assumes that each cluster corresponds to a different Hawkes process and learns cluster-specific excitation patterns from asynchronous event sequences. Compared with standard feature-based clustering methods, TPP-based clustering explicitly models temporal dependencies and excitation structures, leading to more interpretable clusters from a dynamical perspective.

## 8.2 Application in Causal Discovery

In Hawkes process, causal discovery aim to recover the causal structure among difference event types from observed event sequence data. Applications in this category are prevalent in areas like neuroscience, finance, AI for operations, social network, healthcare, and cybersecurity. Here, the focus is not on predicting future events but on uncovering dependencies between event types, often referred to as Granger causality (Granger, 1969). These causal relationships enable better decision-making and provide mechanistic understanding of complex event dynamics. For example, in neuroscience, each neuron can be considered as an event type, and its spike train forms a univariate point process. The spike trains of multiple neurons naturally constitute a multivariate point process. The goal is to determine whether there exists functional connectivity between neurons (Linderman & Adams, 2015; Zhou et al., 2022a). Similarly, in high-frequency financial trading, a large number of asks (sell orders) and bids (buy orders) occur within short periods. Here, all sell orders are treated as one event type, and all buy orders as another. The primary interest lies in understanding the mutual influence between buy and sell orders in the order book (Bacry & Muzy, 2014). In AI operations, TPPs identify system failure root causes by distinguishing primary triggers from secondary effects through their causal structure, guiding prioritized fixes (Cai et al., 2022). Social network analysis similarly leverages Hawkes processes to quantify mutual influence patterns, where user actions (posts, likes, shares) as distinct event types reveal how influential users trigger reaction cascades (Zhou et al., 2013c). Healthcare applications employ TPPs to analyze drug reactions and symptom interactions for improved treatment strategies (Bao et al., 2017). Cybersecurity implementations further demonstrate their value in attack pattern analysis for enhanced defense mechanisms (Bessy-Roland et al., 2021; Fortino et al., 2022).

The modeling framework for these applications in causal discovery builds upon multivariate Hawkes processes (MHP) where a $K$-varaite MHP can be formulated as a collection of $K$ univariate TPPs with the conditional intensity function taking the form of Equation (6). Crucially, we say process $k'$ does not Granger-cause process $k$ if and only if $\phi_{k,k'}(\cdot) = 0$. Therefore, estimating these triggering functions thus directly translates to learning the causal structure. While MLE offers a straightforward solution for learning causal structure, it often produces spurious connections due to finite samples and the lack of sparsity constraints. To tackle this issue, various types of sparsity approaches have been developed, typically categorized into constraint-based and score-based methodologies. Constraint-based approaches address the problem through statistical testing to prune spurious edges. For instance, Runge et al. (2019) proposed a general constraint-based framework for learning causal structure in time series data using conditional independence tests, but it is only applicable to discrete-time processes. Later, Mogensen (2020) proposed a screening algorithm for the Hawkes process, extending it to the continuous-time domain.

In contrast, score-based approaches learn causal structure by optimizing a well-defined criterion with various sparsity regularizations to enforce structural sparsity. For instance, Xu et al. (2016) proposed a nonparametric Hawkes process model with group sparsity regularization, while Zhou et al. (2013c) employed both nuclear norm and $\ell_1$ norm as sparsity regularizations. Idé et al. (2021) used $\ell_0$-regularization via an $\epsilon$-sparsity approach (Phan & Idé, 2019). Alternatively, based on data compression techniques, Jalaldoust et al. (2022) proposed a minimum description length (MDL) criterion for Hawkes processes. Using a similar approach, Hlaváčková-Schindler et al. (2024) introduced a minimum message length criterion, which extends the MDL framework. This extension incorporates prior distributions—such as expert knowledge—over model parameters, enhancing flexibility in structure-related penalization.

By further assuming process stability, Achab et al. (2018) demonstrated that the inference procedure can be accelerated using a cumulant matching strategy based on the analytical form of cumulants (Jovanović et al., 2015), thereby eliminating the need to estimate the triggering function. Recently, several deep point process-based methods have been proposed to move beyond static parametric triggering functions. For example, Zhang & Yan (2021) introduced a variational neural relation inference framework that combines multivariate TPPs with message-passing graphs for probabilistic relation discovery. Yang & Zha (2024) further proposed a variational autoencoder with dynamic latent graphs to capture time-varying dependencies among event types. Wang et al. (2025) modeled multivariate TPPs through neural jump stochastic differential equations with latent graphs, offering a flexible continuous-time framework for relation-aware intensity dynamics. Zhang et al. (2020c) introduced an attribution method to uncover Granger causality, and Wu et al. (2024) explored instance-wise causal structures using Transformer self-attention, aligning the

mechanism with Granger causality principles. These works illustrate that neural causal discovery in TPPs is increasingly shifting toward latent-graph and graph-neural formulations.

Another line of research addresses more realistic scenarios where processes may exhibit nonstationarity, topological dependence, and insufficient temporal resolution. First, nonstationary dynamics frequently emerge when event interactions and background intensities vary over time (Chen et al., 2023). Chen et al. (2022b) further demonstrated that the causal mechanisms can change over time. Second, network effects often exist, where events are influenced not only by their own history but also by their topological neighbors. Failure to account for these dependencies can lead to biased causal estimates. To handle topological dependencies, Cai et al. (2022) proposed the Topological Hawkes Process (THP), which extends temporal convolution to graph-time convolution while employing an EM-based inference approach. Subsequent improvements by Li et al. (2024) enhanced THP's scalability through gradient-based optimization, and Zhu et al. (2024) further generalized the framework using causal-attention Transformers to capture complex network relationships.

For low-resolution scenarios, Trouleau et al. (2021) demonstrated the robustness of cumulant-based methods to observational noise. Later, Cüppers et al. (2024) incorporated delayed effects using a delay-aware MDL criterion to handle observation delays. Furthermore, Qiao et al. (2023) showed that low resolution may lead to instantaneous effects and thus developed a discrete-time structural Hawkes process to handle such instantaneous causal relationships.

## 9 Challenges

Despite the rapid progress of temporal point processes (TPPs), several fundamental challenges remain unresolved. Unlike standard sequence modeling problems, TPPs introduce unique difficulties due to their continuous-time nature, reliance on conditional intensity functions, and strict temporal ordering constraints. These characteristics lead to challenges that go beyond those commonly encountered in general machine learning settings.

**Data and Model**   A major bottleneck for neural TPP research lies not only in the absence of standardized and well-curated benchmarks, but also in the intrinsic heterogeneity of event sequence data. TPP datasets often exhibit irregular time gaps, highly variable sequence lengths, and diverse mark spaces, which makes it difficult to design unified preprocessing pipelines and evaluation protocols across datasets. This heterogeneity is not merely a practical inconvenience; it directly affects model behavior, as different models may implicitly rely on different assumptions about time scales, event density, or mark distributions. Existing datasets differ widely in preprocessing, time resolution, event definitions, and evaluation protocols, which makes reported improvements difficult to compare and often irreproducible. This inconsistency can induce implicit distribution shifts that confound model evaluation and lead to misleading conclusions about architectural superiority. While large-scale studies such as Bosser & Taieb (2023) have taken important steps toward unified evaluation, the field still lacks community-agreed benchmarks with fixed data splits, standardized metrics, and clear task definitions. Some initial efforts have already been made. For example, EasyTPP (Xue et al., 2023a) provides a unified repository that implements a wide range of classical TPP models, ensuring consistency across implementations. Meanwhile, benchmark datasets such as Jiang et al. (2025) further extend evaluation to multimodal and reasoning-oriented tasks. Establishing standardized benchmarks remains particularly challenging in TPPs due to the need to jointly handle temporal structure and event semantics.

**Model Interpretability**   The interpretability gap between classical and neural TPPs is rooted in the role of the conditional intensity function. In traditional models such as Poisson or Hawkes processes, parameters directly correspond to meaningful quantities such as background rates and triggering kernels. In contrast, neural TPPs encode temporal dynamics implicitly in high-dimensional latent states, making it difficult to understand how past events influence future event intensity. This issue is especially critical in applications such as causal discovery, decision support, and scientific modeling, where interpreting event dependencies is often more important than predictive accuracy. While recent works propose attention-based or post-hoc attribution methods, these explanations are often heuristic and lack formal guarantees. A more principled direction is to design neural architectures with built-in inductive biases that preserve interpretable com-

ponents, for example by constraining intensities, kernels, or latent dynamics to have physically meaningful structure. There have also been some initial efforts in this direction. For instance, Meng et al. (2024); Zhou & Yu (2023) attempt to align neural TPPs with traditional statistical TPPs to improve interpretability.

**Model Scalability**   Scalability limitations in TPPs are particularly severe due to the combination of long event sequences and continuous-time modeling requirements. In many real-world applications, event sequences may span tens of thousands of timestamps, while the model must capture dependencies not only across events but also over continuous time intervals. For attention-based models, this leads to quadratic complexity with respect to sequence length. More importantly, unlike standard sequence models, TPPs often require evaluating the conditional intensity function or its integral over time, which introduces additional computational overhead. Although linear-complexity alternatives such as state space models and Mamba (Chang et al., 2024; Gao et al., 2024) have shown promise in sequence modeling, their theoretical properties and inductive biases for point process data remain poorly understood. Future work must go beyond simply replacing attention with efficient modules and instead study how these architectures represent hazard functions, history dependence, and long-term temporal causality.

**Sampling Efficiency**   Sampling efficiency is a particularly critical challenge in TPPs due to the reliance on conditional intensity functions and sequential simulation procedures. Classical sampling methods, such as thinning and inverse transform sampling, require repeated evaluation of the intensity function over time, which becomes computationally expensive for complex neural TPPs. Moreover, unlike discrete sequence models, TPP sampling must ensure temporal validity, such as strictly increasing timestamps and consistency with the underlying intensity function, which further constrains parallelization. As a result, many TPP models are efficient for likelihood evaluation but slow for simulation and forecasting. Recent works have explored alternative generative paradigms to address this issue. For example, flow-based methods (Lüdke et al., 2025; Kerrigan et al., 2024; Shou, 2025), diffusion-based approaches (Yuan et al., 2023; Lüdke et al., 2024; Zhang et al., 2024), and speculative decoding techniques (Gong et al., 2025; Biloš et al., 2025) aim to enable parallel or block-wise generation while preserving temporal consistency. However, these approaches introduce new trade-offs, such as weaker control over conditional structure or higher computational cost.

**Multimodal Modeling**   Real-world event sequences are often accompanied by rich contextual information such as text, images, or sensor data. Integrating such multimodal signals into TPPs presents unique challenges due to the mismatch between continuous-time representations and high-dimensional discrete modalities. In particular, temporal information is structured and continuous, while modalities such as text and images are typically unstructured and discrete, leading to fundamental differences in representation and learning objectives. Current TPPs struggle to integrate such heterogeneous data due to misaligned representations, missing modalities, and high-dimensional inputs. While LLM-based TPPs provide a promising direction by leveraging pretrained multimodal representations, they also introduce challenges in temporal alignment, uncertainty calibration, and controllability. A key open problem is how to jointly model temporal dynamics and multimodal semantics in a principled way, while preserving both statistical consistency and computational efficiency. This direction is still emerging, but initial efforts have already been made to integrate textual and visual information into TPP modeling (Jiang et al., 2025; Liu & Quan, 2024; Zhang et al., 2023; Kong et al., 2025).

## 10   Conclusions

TPPs provide a powerful mathematical framework for modeling asynchronous event sequences across diverse domains such as neuroscience, finance, and social media. Over the years, the field has evolved from traditional parametric models to increasingly flexible nonparametric and neural approaches. In this survey, we reviewed recent progress in three major directions of TPP research: Bayesian TPPs, neural TPPs, and LLM-based TPPs. Each paradigm offers distinct modeling philosophies and strengths—Bayesian methods emphasize uncertainty quantification and principled inference, neural methods prioritize expressive power and scalability, while LLM-based approaches open new avenues for handling complex multimodal sequences. We also provided a taxonomy of representative works, highlighted core modeling principles and estimation

strategies, and discussed ongoing challenges including model interpretability, scalability, and sampling efficiency. In particular, we emphasized recent developments since 2020 that have not been fully covered in earlier surveys, including Bayesian nonparametric models and the emerging trend of applying large language models to TPPs. Looking ahead, we believe that continued progress will hinge on deeper integration across statistical rigor, neural flexibility, and language-model capabilities. Addressing key challenges and enabling broader application of TPPs in real-world, multimodal, and high-resolution settings remain central to advancing the field.

**Acknowledgments**

This work was supported by the NSFC Projects (No.62576346, U24A20233, 62406080), the MOE Project of Key Research Institute of Humanities and Social Sciences (22JJD110001), the CCF-DiDi GAIA Collaborative Research Funds (CCF-DiDi GAIA 202521), the fundamental research funds for the central universities, and the research funds of Renmin University of China (24XNKJ13), and Beijing Advanced Innovation Center for Future Blockchain and Privacy Computing.

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
