# OpenReview forum: "Advances in Temporal Point Processes: Bayesian, Neural, and LLM Approaches"
_TMLR — Accepted by TMLR_

### Review · Reviewer_4sLi · 2026-03-03

**Summary Of Contributions:**

This paper provides a comprehensive survey of Temporal Point Processes (TPPs), organizing the field's evolution into three primary paradigms: Bayesian Approaches (with a focus on nonparametrics), Neural TPPs (including RNN, Transformer, and ODE-based models), and the emerging field of LLM-based TPPs. It extends the scope of previous surveys by incorporating advancements in neural TPPs post-2021 and LLM-based approaches.

## Strengths

1. The inclusion of research up to 2025—specifically the "Neural TPP" developments from the last three years and the emerging "LLM-TPP"—ensures the survey addresses the most current frontier of the field.

2. Besides listing approaches in each category, the authors offer an analysis of the advantages and disadvantages of each model category.

3. The illustrations—particularly Figures 2, 3, and 4—significantly enhance the paper’s accessibility.

## Weaknesses

1. While the inclusion of LLM-based TPPs is a major selling point, the discussion is currently somewhat lean compared to the rigorous treatment of the Bayesian and Neural sections. To mitigate the "newness" of the field, the authors could strengthen this section by discussing adjacent domains.

2. Section 6 (Model Training) relies purely on texts. The absence of key equations and summary figures makes it difficult for readers to get the differences of varying estimators.

3. While the paper mentions data challenges, a dedicated section for existing data and benchmarks is recommended. This should detail dataset sources, statistics (event counts/types), evaluation tasks (prediction vs. simulation), and standard metrics. Given the community's struggle with fragmented benchmarks, data and evaluation are as important as models for the community.

4. Additional tasks, such as clustering[1], can be discussed in Section 7 (Applications).

[1] A dirichlet mixture model of hawkes processes for event sequence clustering.

**Audience:**

Yes

**Audience Explanation:**

This survey is valuable for readers new to the field as it provides a self-contained introduction to fundamental concepts. Furthermore, it serves individuals seeking cross-field research by bridging the gap between traditional statistical modeling and modern deep learning paradigms like Neural TPPs and LLMs.

**Broader Impact Concerns:**

No ethical implications need to be highlighted here.

**Claims And Evidence:**

Yes

**Claims Explanation:**

The authors provide a comprehensive summary of the TPP landscape. Technical descriptions of model families—spanning Bayesian nonparametrics, neural architectures, and emerging LLM integrations—are consistently supported by formal equations and figures (notably Figures 2, 3, and 4) that clarify structural differences.

**Requested Changes:**

1. Add equations or illustrations to explain differences in training objectives in Section 6.
2. Add a dedicated section for existing data and benchmarks as suggested in Weakness 3.
3. The following related works can be included: [1] proposes a Transformer-based model without setting the form of intensity for flexible modeling; [2]~[4] are related to causal discovery using neural TPPs.

[1] Transformer Embeddings of Irregularly Spaced Events and Their Participants.

[2] Learning Neural Jump Stochastic Differential Equations with Latent Graph for Multivariate Temporal Point Processes.

[3] Neural Relation Inference for Multi-dimensional Temporal Point Processes via Message Passing Graph.

[4] A Variational Autoencoder for Neural Temporal Point Processes with Dynamic
Latent Graphs.

---

> ### Author Response · Authors · 2026-03-16
> **Response**
>
> We thank the reviewer for the detailed suggestions on improving the clarity and coverage of the survey.
>
> >Q1: The discussion of LLM-based TPPs is relatively limited.
>
> R1: We expanded this section by discussing connections between LLM-based approaches and related research areas such as sequential foundation models, temporal representation learning, and retrieval-based modeling of event sequences. This provides broader context for the emerging role of LLMs in event sequence modeling.
>
> **Revision location: The first paragraph in Section 5, and we add a separate paragraph “On the boundary of the TPP framework”at the end of Section 5**
>
> >Q2: Section 6 lacks equations explaining the differences between training objectives.
>
> R2:We added key equations for the main training objectives discussed in this section, including MLE, Wasserstein distance objectives, noise contrastive estimation (NCE), and Fisher divergence (score matching). These additions clarify the differences between estimation strategies.
>
> **Revision location: Section 7**
>
> >Q3: A dedicated section on data and benchmarks is recommended.
>
> R3: We agree and added a new section summarizing datasets, benchmarks, and evaluation protocols used in TPP research.
>
> **Revision location: Section 6**
>
> >Q4: Additional tasks such as clustering could be discussed.
>
> R4: We added a paragraph on event sequence clustering, including the Dirichlet mixture model of Hawkes processes, highlighting that TPPs can also be used to discover latent structure in heterogeneous event streams.
>
> **Revision location: the end of Section 8.1**
>
> >Q5: Include additional related works.
>
> R5: We incorporated the suggested references, including works on Transformer-based event embeddings and neural TPP models for causal discovery using latent graph structures, and discussed them in the relevant sections.
>
> **Revision location: Sections 4.2 and 8.2**

---

### Review · Reviewer_ahWk · 2026-03-08

**Summary Of Contributions:**

This paper makes a comprehensive survey of TPP model works up to late 2025. This TPP survey covers the following topics including: fundamentals of TPP, Bayesian non-parametric TPP, neural TPP, LLM-based TPP, model training, and applications. As a relatively new TPP survey work, it not only provides a comprehensive and modern update on neural TPP and Bayesian non-parametric TPP compared to existing surveys. The coverage of Bayesian non-parametric TPP was often neglected by previous surveys. It also expands to a systematic survey of LLM-based TPP approaches and approaches based on multi-modal data in this rapidly changing domain. The survey is well organized and presented in a clean style. Specifically, it also presents comparisons across different methods using tables on several topics, which is easy to follow. However, the survey doesn't cover benchmark datasets and evaluation metrics, which are also undergoing changes in recent development of TPP research. In addition, some latest family of approaches like diffusion-based TPP approaches are not included as a separate topic for literature review.

**Audience:**

Yes

**Audience Explanation:**

As a survey paper, the work does not necessarily provide new findings. However, future event prediction and temporal point process modelling are research topics many AI researchers are interested in. As the work provides a modern and comprehensive survey on the latest progress of TPP research, the event sequence modelling and TPP research communities are potential audiences of this work.

**Broader Impact Concerns:**

sAs a survey paper on existing methods, I don't think this work is required to have a Broader Impact Statement, but I do encourage the author to take a look at the second point of suggested changes.

**Claims And Evidence:**

Yes

**Claims Explanation:**

The work claims the following major difference from existing survey works:
1. Inclusion of LLM-based TPP approaches. This claim is supported by Section 5 as a standalone section.
2. Focus on Bayesian non-parametric methods, which is overlooked by previous surveys. This claim is supported by Section 3.
3. It also includes a few important latest works that are not included in previous surveys, like Neural Jump-Diffusion TPPs, Transformer Hawkes Process, Neural Hawkes Process, and AttNHP.

**Requested Changes:**

I would like to request the author to include a section on benchmark datasets and evaluation metrics to the survey of this work. New datasets like Amazon [1] and Taobao [2] have become available for evaluating TPP models at a larger scale in recent years. Evaluation metrics focusing on long-horizon prediction [2] have been proposed, and efforts to establish standard benchmarks [3] are also appearing. The inclusion of benchmark datasets and evaluation metrics is crucial to help an external audience or new AI researchers interested in TPP understand and get started with this research topic.

I would also suggest the author consider the following changes:
1. Include a section on diffusion-based approaches for TPP. Many diffusion-based TPP works are already included in the reference, and I would suggest the author consider including the following work as well:
* Lüdke, David, et al. Add and thin: Diffusion for temporal point processes. NeurIPS 2023

2. Some of the tasks studied by the works in the extension of LLM-based approaches like retrieval of event sequences and QA tasks in DanmakuTPPBench does not necessarily fits in the traditional framework of temporal point process as a stochastic process. This could be a good opportunity for the authors to discuss the limitations of the TPP framework for handling event sequence data.

References:

[1] Ni, J. Amazon review data, 2018.

[2] Xue, S, et al. Hypro: A hybridly normalized probabilistic model for long-horizon prediction of event sequences. NeurIPS 2022

[3] Xue, Siqiao, et al. Easytpp: Towards open benchmarking temporal point processes. ICLR 2024

---

> ### Author Response · Authors · 2026-03-16
> **Response**
>
> We thank the reviewer for the helpful suggestions on benchmarks, diffusion-based approaches, and the scope of LLM-based TPP research. These comments have helped improve the completeness of the survey.
>
> >Q1：Include a section on benchmark datasets and evaluation metrics.
>
> R1：We agree that datasets and evaluation protocols are essential for guiding new researchers in the TPP community. In the revised manuscript, we added a new subsection “Datasets, Benchmarks, and Evaluation Metrics”, summarizing commonly used datasets, benchmark efforts such as EasyTPP, and standard evaluation tasks and metrics.
>
> **Revision location: Section 6**
>
> >Q2：Include a section on diffusion-based approaches for TPP.
>
> R2：Following the reviewer’s suggestion, we added a new subsection “Diffusion-based TPPs”, which reviews recent work introducing diffusion generative models into temporal point processes.
>
> **Revision location: Section 4.4**
>
> >Q3：Some LLM-based tasks may not fit the traditional stochastic process framework of TPPs.
>
> R3：We agree with this observation. In the revised manuscript, we added a discussion clarifying that classical TPP models focus on continuous-time event occurrence modeling, whereas some LLM-based benchmarks involve tasks such as retrieval and question answering over event sequences. We highlight that these tasks extend TPP research toward semantic reasoning over temporally structured data, while also discussing the limitations of the classical TPP framework.
>
> **Revision location: The first paragraph in Section 5, and we add a separate paragraph “On the boundary of the TPP framework”at the end of Section 5**

---

### Review · Reviewer_wQbk · 2026-04-13

**Summary Of Contributions:**

This work presents a survey of methods for Temporal Point Processes (TPPs). It opens with a background section defining the current ways to model TPPs, followed by a survey of methods organized into three families: Bayesian Nonparametric TPPs, Neural TPPs, and LLM-based TPPs. More technical aspects — datasets, benchmarks, evaluation protocols, and training procedures — are then detailed. The paper closes with applications and a list of open challenges.

**Strengths**

- The contribution is real: the most recent existing survey dates from 2021 and overlooks recent advances, notably those involving LLMs.
- The pedagogical value is very high. As someone not working in this specific field, I found the paper clear and didactic. I particularly appreciated Figures 2 to 6.
- The paper is extensive and well organized, covering a wide range of aspects.

**Weaknesses**

- Little information is given about the methodology used to gather the surveyed literature.
- One of the main additions compared to previous surveys is the inclusion of new method families, yet some of them — notably diffusion-based TPPs — are only lightly discussed.
- While the advantages and disadvantages of methods are well presented, the survey offers limited discussion of their empirical performance on benchmarks.

**Audience:**

Yes

**Audience Explanation:**

As noted in the summary, the contribution is real: the most recent existing survey dates from 2021 and overlooks recent advances, notably those involving LLMs. This alone sufficiently motivates the survey and ensures its relevance to part of TMLR's audience.

**Claims And Evidence:**

Yes

**Claims Explanation:**

Overall, the manuscript is very solid and, in my opinion, contains all the content expected of a survey. The technical details are well presented and, as far as I could observe, free of errors, and the body of surveyed work is sufficient to constitute a comprehensive study.

That said, I want to point out that, not being an expert in this specific field, it is entirely possible that I missed some important works.

**Requested Changes:**

In its current state, I have few requests concerning the manuscript. However, here is a short list of changes that I believe could enhance its quality:
- Provide more details about the procedure used to conduct the survey: tools used, scope of the search, inclusion criteria, etc.
- Extend Section 4.4. In particular, the disadvantages of non-autoregressive sequence generation are unclear to me as currently presented.
- Add, if possible, more insights into how the surveyed methods perform on the benchmarks mentioned.
- The Challenges section could, in my opinion, be improved. As it stands, it feels somewhat generic: benchmarking, interpretability, scalability, and efficiency are common issues across most signal processing fields, and the current discussion offers limited insight specific to TPPs.

---

> ### Author Response · Authors · 2026-04-16
> **Response**
>
> We thank the reviewer for the constructive and insightful feedback. We have carefully revised the manuscript to address all the concerns. Below we summarize the key changes.
>
> > Q1: Survey methodology clarification
>
> R1: We agree that the original manuscript did not explicitly describe the literature collection procedure. To address this, we have added a new paragraph *“Survey Methodology”* in the Introduction.
>
> Specifically, we now clarify:
> - the scope of the literature search (major ML/statistics venues such as NeurIPS, ICML, ICLR, AISTATS, JMLR, etc.),
> - the time span covered (primarily 2020–2025 with inclusion of foundational works),
> - the inclusion criteria (model, inference, and application contributions relevant to TPPs),
> - and the overall taxonomy used in the survey.
>
> > Q2: Clarification of non-autoregressive (diffusion-based) limitations
>
> R2: We appreciate this suggestion. In Section 4.4 (Diffusion-based TPPs), we have substantially expanded the discussion on the limitations of non-autoregressive sequence generation.
>
> In particular, we now explicitly explain:
> - the lack of explicit conditional structure compared to autoregressive TPPs,
> - the difficulty in enforcing temporal consistency and causal dependence,
> - the increased computational cost due to iterative denoising,
> - and challenges in likelihood-based evaluation and fine-grained temporal modeling.
>
> > Q3: Empirical insights on benchmarks
>
> R3: We agree that the original manuscript focused more on datasets and evaluation protocols than on summarizing empirical findings.
>
> To address this, we have added a new paragraph *“Empirical Insights”* at the end of Section 6. This paragraph synthesizes consistent observations across prior works, including:
> - the strong performance of neural (especially Transformer-based) TPPs on next-event prediction,
> - the efficiency advantages of density-based parameterizations,
> - the challenges of long-horizon prediction across all model classes,
> - and the task-dependent advantages of LLM-based and multimodal TPPs.
>
> > Q4: Strengthening the Challenges section with TPP-specific insights
>
> R4: We thank the reviewer for pointing out that the original Challenges section was too generic. We have significantly revised this section to emphasize challenges that are specific to TPPs.
>
> The revised version now explicitly highlights:
> - the impact of continuous-time modeling and irregular event structure in data,
> - the role of conditional intensity functions in interpretability and model design,
> - the additional computational burden of intensity evaluation and integration in scalability,
> - the inherent sequential constraints of sampling due to temporal ordering,
> - and the mismatch between continuous-time dynamics and discrete multimodal representations.
>
> Importantly, we retained the original structure and references, while incorporating these TPP-specific perspectives throughout the section.

---

> > ### Comment · Reviewer_wQbk · 2026-04-16
> > **Answer to the revision of the manuscript**
> >
> > I would like to thank the authors for their response. On my end, I consider that all my concerns have been addressed and have no further comments.

---

### Decision · Action_Editor_LwaR · 2026-06-03

**Recommendation:** Accept as is

**Audience:**

Yes

**Audience Explanation:**

TPP and event-sequence modeling are of active interest to the ML community, and the most recent comparable survey (2021) misses recent LLM-based and multimodal developments. This work fills a timely gap and serves both newcomers and cross-field researchers. All three reviewers confirmed its relevance.

**Claims And Evidence:**

Yes

**Claims Explanation:**

All three reviewers agreed the claims are well supported. The technical descriptions of the surveyed model families are clear and free of evident errors, and the stated contributions over prior surveys (LLM-based, Bayesian nonparametric, and recent neural TPP methods) are each backed by corresponding sections. Revisions further strengthened this with added methodology, benchmark, and training-objective details.